# Avoid What You Know: Divergent Trajectory Balance for GFlowNets

Pedro Dall'Antonia [1]   Tiago da Silva [2]   Daniel Csillag [1]   Salem Lahlou [2]   Diego Mesquita [1]

## Abstract

Generative Flow Networks (GFlowNets) are a flexible family of amortized samplers trained to generate discrete and compositional objects with probability proportional to a reward function, learning a policy function over an intractably large state graph by minimizing a stochastic objective over sampled trajectories. However, learning efficiency is constrained by the model's ability to rapidly explore diverse high-probability regions during training. To mitigate this issue, recent works have focused on incentivizing the exploration of unvisited and valuable states via curiosity-driven search and self-supervised random network distillation, which tend to waste samples on already well-approximated regions of the state space. In this context, we propose *Adaptive Complementary Exploration* (ACE), a principled algorithm for the effective exploration of novel and high-probability regions when learning GFlowNets. To achieve this, ACE introduces an *exploration* GFlowNet explicitly trained to search for high-reward states in regions underexplored by the *canonical* GFlowNet, which learns to sample from the target distribution. Through extensive experiments, we show that ACE consistently and significantly improves upon prior work in terms of approximation accuracy to the target distribution and discovery rate of diverse high-reward states.

## 1. Introduction

Generative Flow Networks (GFlowNets; Bengio et al., 2021) are powerful reward-driven generative models designed to sample from distributions over compositional objects (e.g., graphs and sequences), with a range of applications in scientific discovery (Wang et al., 2023), combinatorial optimization (Zhang et al., 2023a;b), and

approximate inference (Malkin et al., 2023). Building on the compositional structure of the target distribution's support, GFlowNets create valid samples by starting from an initial state and iteratively drawing from a forward policy. Learning a GFlowNet then boils down to finding policies that satisfy a set of identities called *balance conditions*, which ensure sampling correctness.

To achieve this, we train GFlowNets by minimizing the logarithmic residuals of a balance condition over the state graph (Malkin et al., 2022; Tiapkin et al., 2024). Nonetheless, since analytically sweeping through the entire state graph is intractable, we instead average over the residuals in a set of sampled trajectories. Conventionally, these trajectories are drawn from an $\epsilon$-greedy *exploratory policy* consisting of a mixture of the forward and an uniform policy at each time step (Bengio et al., 2023). In theory, the uniform component provides full support to the sampling distribution, preventing mode collapse. In practice, however, uniform search might not be enough to warrant the exploration of multiple high-probability regions if the target reward is sparsely distributed (Malkin et al., 2023; Shen et al., 2023).

To improve exploration and provide better support coverage, recent works considered learning the exploratory policy alongside the GFlowNet. Inspired by curiosity-driven exploration, Kim et al. (2025c) recently suggested training such a policy to sample from a log-linear mixture of the GFlowNet's loss and the reward function. Concurrently, Madan et al. (2025) proposed instead targeting a combination of the original (extrinsic) reward and an intrinsic reward based on self-supervised random network distillation (RND). In both cases, exploration is guided by a GFlowNet steered by the loss function of an external deep neural network—either that of the underlying sampler (Kim et al., 2025c; Malek et al., 2026) or the RND loss (Madan et al., 2025). To distinguish the learned exploratory policy from the GFlowNet trained to sample from the target distribution, we will refer to the latter as the *canonical* GFlowNet.

From a fundamental viewpoint, the crux of designing an appropriate exploratory policy is ensuring it samples trajectories from high-reward but underexplored regions during training. In this work, we cast this problem as satisfying a new balance condition for exploration, which we call *divergent trajectory balance* (DTB). In a nutshell, DTB enforces

---

[1]School of Applied Mathematics, Getulio Vargas Foundation [2]MBZUAI. Correspondence to: Pedro Dall'Antonia <pepedall1997@gmail.com>.

*Proceedings of the $43^{rd}$ International Conference on Machine Learning*, Seoul, South Korea. PMLR 306, 2026. Copyright 2026 by the author(s).

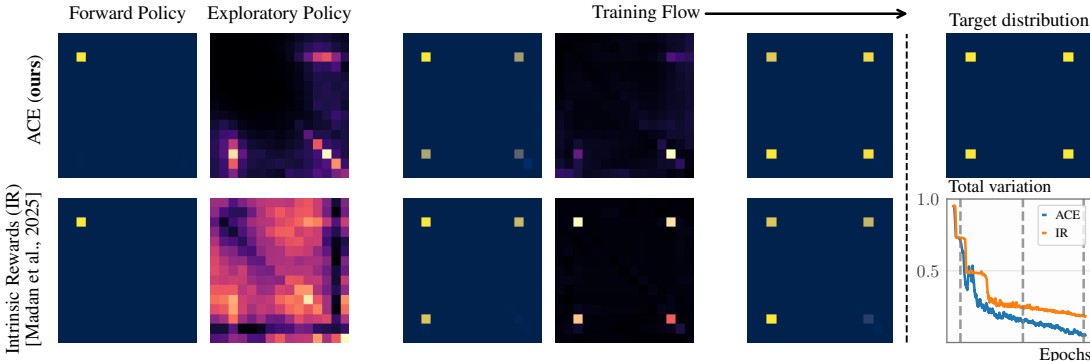

*Figure 1.* When learning the exploratory policy based on a combination of intrinsic and extrinsic rewards—see Equations (6) and (7)—, the model may overemphasize well-learned states (bottom row). In contrast, ACE *avoids* sampling trajectories from over-explored regions of the state space by design (top row), which improves mode discovery and accelerates learning convergence (rightmost panel) to the target. This figure shows the marginal distribution of forward and exploratory policies at different training points (marked as dashed vertical lines).

standard trajectory balance on under-sampled regions while imposing zero probability to trajectories terminating in over-sampled states. As a consequence, any exploratory policy satisfying DTB concentrates its sampling on high-reward terminal states under-represented by the canonical GFlowNet.

The ensuing algorithm, called *Adaptive Complementary Exploration* (ACE), can be interpreted as a two-player game: the exploratory policy is continually adjusted to sample from under-sampled areas, and the GFlowNet being trained is updated using trajectories generated by the exploratory policy and its own policy. In doing so, we prevent the canonical GFlowNet from repeatedly visiting only a small number of high-reward subsets of the state space, while ignoring others, a behavior known to slow down training (Vemgal et al., 2023; Atanackovic & Bengio, 2024). We also characterize the system's equilibrium in Propositions 3.4 and 3.10.

When viewed through the lens of the flow network analogy, which inspired the development of GFlowNets (Bengio et al., 2021), ACE implements a principled mechanism for transferring excess probability mass from over- to under-allocated nodes. We illustrate this in the diagram of Figure 1. Notably, our experiments on peptide discovery, bit sequence design, combinatorial optimization, and more, confirm ACE significantly speeds up training convergence and the discovery of diverse, high-reward states when compared against prior techniques for improved GFlowNet exploration.

In summary, our contributions are the following.

1. We propose *Adaptive Complementary Exploration* (ACE), a new method for effective state space exploration in GFlowNet training. Based on the novel *divergent trajectory balance* (DTB), ACE focuses on generating trajectories from high-reward underexplored regions by the canonical GFlowNet, increasing sample diversity and mitigating mass concentration on a few modes.

2. We show ACE prevents distributional collapse of the

canonical GFlowNet (Proposition 3.9) by biasing sampling towards infrequently visited but valuable regions of the state space (Propositions 3.4 and 3.10).

3. We evaluate our method on diverse and standard benchmark tasks, including the grid world, peptide discovery, and bit sequence generation, and show that it often results in drastically faster identification of diverse and high-reward states relatively to prior approaches.

## 2. Preliminaries & Related Works

**Definitions.** Our objective is to sample objects $x$ from a discrete space $\mathcal{X}$ in proportion to a *reward* function $R : \mathcal{X} \to \mathbb{R}_+$. We say $\mathcal{X}$ is *compositional* if there is a directed acyclic graph $G = (\mathcal{S} \cup \mathcal{X}, \mathcal{E})$ over an extension $\mathcal{S} \cup \mathcal{X}$ of $\mathcal{X}$ with edges $\mathcal{E}$ having the following properties.

1. There exists a unique $s_o \in \mathcal{S}$ s.t. (i) $s_o$ is connected to any $s \in \mathcal{S}$ via a directed path, denoted $s_o \rightsquigarrow s$, and (ii) $s_o$ has no incoming edges. We call $s_o$ the *initial state*.

2. There exists a unique $s_f \in \mathcal{S}$ s.t. (i) $s \to s_f \in \mathcal{E}$ if and only if $s \in \mathcal{X}$ and (ii) $s_f$ has no outgoing edges. We refer $s_f$ as the *final state* and to $\mathcal{X}$ as the set of *terminal states*.

Under these conditions, we refer to $G$ as a *state graph*. Illustratively, let $\mathcal{X}$ be the space of $k$-sized multisets with elements selected from $\{1, \ldots, n\}$. Then, we construct $G$ by defining $s_o = \emptyset$ and $\mathcal{S}$ as the space of multisets with size up to $k - 1$, including a directed edge from $s$ to $s'$ if and only if $s'$ differs from $s$ by a single additional element. Importantly, although $|\mathcal{X}| = \mathcal{O}(n^k)$ is combinatorially explosive, each $s \in \mathcal{S}$ has exactly $n$ children in $G$.

A GFlowNet is an amortized sampler defined over $G$. Section 2.1 reviews the formalism behind GFlowNets, and Section 2.2 discusses the limitations of prior art addressing the problem of inefficient exploration in GFlowNet training.

## 2.1. GFlowNets

To start with, we recast the problem of directly sampling from $R$ on $\mathcal{X}$ to that of learning an *amortized policy function* $p_F \colon \mathcal{S} \times (\mathcal{S} \cup \mathcal{X}) \to [0, 1]$ such that $p_F(s, \cdot)$ is a probability measure supported on the children of $s$ in $G$. To achieve this goal, we parameterize $p_F(s, \cdot)$ as a softmax deep neural network trained to satisfy

$$p_\top(x) := \sum_{\tau \colon s_o \rightsquigarrow x} \prod_{(s,s') \in \tau} p_F(s, s') \propto R(x), \qquad (1)$$

in which the sum covers all trajectories from $s_o$ to $x$ in $G$. We refer to $p_F$ as the *forward policy* and $p_\top(\cdot)$ as its induced marginal distribution over $\mathcal{X}$. For conciseness, we will often omit $s_o$ and write $p_F(\tau) := \prod_{(s,s') \in \tau} p_F(s, s')$ as the forward probability of a trajectory $\tau$ in $G$. As exact computation of $p_\top$ is intractable, we introduce a *backward* policy $p_B \colon (\mathcal{S} \cup \mathcal{X}) \times \mathcal{S} \to [0, 1]$ on the transposed state graph $G^\mathsf{T}$, and jointly search for $p_F$ and $p_B$ satisfying the *trajectory balance* (TB) condition for a learned constant $Z$,

$$Z \cdot p_F(\tau) = p_B(\tau|x) \cdot R(x), \qquad (2)$$

in which $p_B(\tau|x) = \prod_{(s,s') \in \tau} p_B(s', s)$ is the backward probability of $\tau$. As shown by Malkin et al. (2022); Madan et al. (2022), this can be achieved by solving the following stochastic program over $Z$ and policies $p_F$ and $p_B$,

$$\min_{Z, p_F, p_B} \mathbb{E}_{\tau \sim p_E} \left[ \left( \log \frac{Z \cdot p_F(\tau)}{p_B(\tau|x) R(x)} \right)^2 \right], \qquad (3)$$

in which $p_E$ is an *exploratory* policy fully supported on the trajectories in $G$. Other loss functions, e.g., sub-trajectory balance (Madan et al., 2022) and detailed balance (Bengio et al., 2023), have also been studied. We refer the reader to Viviano et al. (2025) for a modular implementation of these objectives and standard benchmarks. By letting $\theta$ be the parameters of our models for $p_F$, $p_B$ and $Z_\theta$, we define

$$\mathcal{L}_{\mathrm{TB}}(\theta; \tau) = \left( \log \frac{Z_\theta \cdot p_F(\tau; \theta)}{p_B(\tau|x; \theta) R(x)} \right)^2 \qquad (4)$$

If context is clear, we will often exclude $\theta$ from the notations of $p_F$ and $p_B$ to avoid notational clutter. Also, notice that a GFlowNet induces a reward function s.t., for each $x \in \mathcal{X}$,

$$\hat{R}_\theta(x) = Z_\theta \cdot p_\top(x) = \mathbb{E}_{\tau \sim p_B(\cdot|x)} \left[ Z_\theta \cdot \frac{p_F(\tau; \theta)}{p_B(\tau|x; \theta)} \right]. \quad (5)$$

In particular, when $\mathcal{L}_{\mathrm{TB}}(\theta; \tau) = 0$ for all $\tau$, the induced reward $\hat{R}_\theta(x)$ matches the true reward $R(x)$ for each $x \in \mathcal{X}$.

## 2.2. Learning GFlowNets

The choice of $p_E$ is paramount for the effective training of GFlowNets (Bengio et al., 2021; Kim et al., 2025a).

Traditionally, $p_E$ is defined as an $\epsilon$-*greedy* version of $p_F$, $p_E(s, \cdot) = (1 - \epsilon) p_F(s, \cdot) + \epsilon p_U(s, \cdot)$, in which $p_U(s, \cdot)$ is an uniform distribution over the children of $s$ in the state graph. For multi-modal target distributions, however, an $\epsilon$-greedy policy might struggle to visit certain modes when $p_F$ is near-collapsed into a subset of the high-probability regions, as both $p_U$ and $p_F$ would then assign negligible probability to the unvisited high-reward states during training. We illustrate this phenomenon in Figure 2. To address this limitation, recent research has focused on the design of sophisticated metaheuristics (Kim et al., 2024; Boussif et al., 2024), reward shaping (Pan et al., 2023; Jang et al., 2024), and ad hoc techniques (Rector-Brooks et al., 2023; Hu et al., 2025) for enhanced state graph exploration, which can be used complementarily to our method.

Additionally, there is a growing body of literature showcasing the effectiveness of parameterizing $p_E$ as an *exploration* GFlowNet trained to sample from a novelty-promoting modification of $R$. Inspired by the literature of curiosity-driven learning, for instance, Kim et al. (2025c) suggested training $p_E$ to sample from a weighted average between $R(x)$ and the loss function $\mathcal{L}_{\mathrm{TB}}$ in Equation (4) in log-space, that is,

$$\log R_{\mathrm{AT}}(x) = \log R_{\mathrm{T}}(x) + \alpha \log R(x),$$

with $\alpha > 0$ and, defining $\delta(\tau) := \log R(x) + \log p_B(\tau|x) - \log p_F(\tau) - \log Z$ as the base residual in Equation (3),

$$\log R_{\mathrm{T}}(x) = \mathbb{E}_{\tau \sim p_B(\cdot|x)} \left[ \log \left( \epsilon + \left( 1 + C \cdot \mathbf{1}_{\delta(\tau) > 0} \right) \delta(\tau)^2 \right) \right]. \tag{6}$$

We refer to this class of methods as Adaptive Teachers (AT) GFlowNets throughout this work. Similarly, Madan et al. (2025) proposed introducing *intrinsic rewards* based on random network distillation (RND; Burda et al., 2018) as a proxy for novelty when learning $p_E$, resulting in

$$R_{\mathrm{SA}}(x; \tau) = \left( R(x)^{\beta_1} + \left( \sum_{s \in \tau} \mathrm{R}_{\mathrm{A}}(s) \right)^{\beta_2} \right)^{\beta_3}, \quad (7)$$

in which $\beta_1, \beta_2, \beta_3 > 0$ are positive constants and $R_{\mathrm{A}}(s) = \|\psi(s) - \psi_{\mathrm{random}}(s)\|^2$ is based on RND of a neural network $\psi$ into a randomly fixed model $\psi_{\mathrm{random}}$ (Pan et al., 2023). The reader should notice that the reward function $R_{\mathrm{SA}}$ for $p_E$ is path-dependent. As in Madan et al. (2025), we call this approach Sibling Augmented (SA) GFlowNets.

From an empirical standpoint, however, $R_{\mathrm{T}}$ and $R_{\mathrm{A}}$ are incomparable to $R$. The reason for this is that while $R$ often represents a physical quantity, such as binding affinity of drugs (Bengio et al., 2021) or a Bayesian posterior (Malkin et al., 2023), $R_{\mathrm{T}}$ and $R_{\mathrm{A}}$ are simply error functions of neural networks designed to capture novelty *indirectly*. Hence, the mechanisms by which either method addresses the problem of insufficient exploration of high-reward

regions that hampers GFlowNet training remain elusive. We defer a comprehensive discussion of related works to Section B in the supplement.

Our method, presented in the next section, circumvents this issue by *directly* promoting the visitation of novel states through the novel *divergent trajectory balance* (DTB) loss.

## 3. Adaptive Complementary Exploration

**Divergent Trajectory Balance.** During training, the canonical GFlowNet may over-allocate probability mass to certain trajectories, hampering exploration of novel and high-reward regions. We illustrate this in Figure 2. When trained to sample from the RINGS distribution (see Section 4) via $\epsilon$-greedy exploration, a GFlowNet concentrates most of its probability in a single high-reward region of the state space, under-representing the second, more distant mode. To formalize this intuition, we define the set of over-sampled trajectories below. For clarity, we will henceforth refer to a GFlowNet as $\mathfrak{g} = (Z, p_F, p_B)$.

**Definition 3.1** (Over- & Under-Allocated regions). Let $\mathfrak{g} = (Z, p_F, p_B)$ be a GFlowNet and $\alpha > 0$. We define the set of *over-allocated states* with respect to $\alpha$ as

$$\mathrm{OA}(\alpha, \mathfrak{g}) = \{x \in \mathcal{X} \colon \hat{R}_{\mathfrak{g}}(x) \geq \alpha \cdot R(x)\},$$

in which $\hat{R}_{\mathfrak{g}}$ is the GFlowNet's induced reward function described in Equation (5). Similarly, we define

$$\mathrm{UA}(\alpha, \mathfrak{g}) = \mathcal{X} \setminus \mathrm{OA}(\alpha, \mathfrak{g})$$

as the set of states with *under-allocated* probability mass. With a slight abuse of notation, we write $\tau \in \mathrm{OA}(\alpha, \mathfrak{g})$ (resp. $\tau \in \mathrm{UA}(\alpha, \mathfrak{g})$) to indicate that $\tau$ starts at $s_o$ and finishes at some $x \in \mathrm{OA}(\alpha, \mathfrak{g})$ (resp. $x \in \mathrm{UA}(\alpha, \mathfrak{g})$). When $\mathfrak{g}$ is clear, we will simply write $\mathrm{OA}(\alpha)$ and $\mathrm{UA}(\alpha)$.

Our objective is to learn an exploratory policy sampling high-reward states in $\mathrm{UA}(\alpha)$ while avoiding trajectories in $\mathrm{OA}(\alpha)$. This can be achieved by enforcing the *DTB* condition (Definition 3.2). Given the terminology, we denote the exploration GFlowNet as $\mathfrak{g}_{\nabla} = (Z_{\nabla}, p_F^{\nabla}, p_B^{\nabla})$.

**Definition 3.2** (DTB). Let $\mathfrak{g}$ and $\mathfrak{g}_{\nabla}$ be GFlowNets. We define the *divergent trajectory balance* (DTB) of $\mathfrak{g}_{\nabla}$ *with respect to* $\mathfrak{g}$ for a threshold $\alpha > 0$ and exponent $\beta > 0$ as

$$Z_{\nabla} \cdot p_F^{\nabla}(\tau) = R(x)^{\beta} \cdot p_B^{\nabla}(\tau|x) \text{ if } \tau \in \mathrm{UA}(\alpha, \mathfrak{g}),$$
$$Z_{\nabla} \cdot p_F^{\nabla}(\tau) = 0 \text{ otherwise.}$$

As in Definition 3.1, we will often omit $\alpha$, $\beta$, and $\mathfrak{g}$ when referring to the DTB of $\mathfrak{g}_{\nabla}$. Similarly to prior work (Madan et al., 2025; Kim et al., 2025c), we train the exploration GFlowNet on a tempered reward function to facilitate state space navigation, a technique that has been empirically

shown to be effective (Zhou et al., 2023) and is rooted in the literature of simulated annealing for Markov chain Monte Carlo (Kirkpatrick et al., 1983). Intuitively, the DTB prunes the support of $p_F^{\nabla}$ to the set of trajectories with under-allocated probability mass by the canonical GFlowNet ($\mathfrak{g}$). The value of $\alpha$ dictates how much of $p_F^{\nabla}$'s support is trimmed, with larger values corresponding to larger supports (i.e., $\alpha \mapsto \mathrm{UA}(\alpha)$ is increasing w.r.t. set inclusion).

In order to learn an exploration GFlowNet $\mathfrak{g}_{\nabla}$ abiding by the conditions in Definition 3.2, we design a loss function we can optimize via gradient descent. Towards this goal, we notice that the DTB conditions can be rewritten as

$$Z_{\phi}^{\nabla} \, p_F^{\nabla}(\tau; \phi) = R(x) \, p_B^{\nabla}(\tau; \phi) \, \mathbb{I}[\tau \in \mathrm{UA}(\alpha, \mathfrak{g})], \quad \forall \tau;$$

equivalently, dividing by $R(x) \, p_B^{\nabla}(\tau; \phi)$ and adding $\mathbb{I}[\tau \in \mathrm{OA}(\alpha, \mathfrak{g})]$ on both sides (recall that $\mathrm{OA}(\alpha) \cup \mathrm{UA}(\alpha) \coloneqq \mathcal{X}$),

$$\frac{Z_{\phi}^{\nabla} \, p_F^{\nabla}(\tau; \phi)}{R(x) \, p_B^{\nabla}(\tau; \phi)} + \mathbb{I}[\tau \in \mathrm{OA}(\alpha, \mathfrak{g})] = 1, \quad \forall \tau. \quad (8)$$

Drawing on this, we define the *divergent trajectory balance loss* $\mathcal{L}_{\nabla}$ as the log-squared residual between the left- and right-hand sides of Equation (8)—analogously to the TB (Malkin et al., 2022) and SubTB (Madan et al., 2022) losses.

**Definition 3.3** (DTB Loss). Let $\mathfrak{g}$ and $\mathfrak{g}_{\nabla}$ be GFlowNets. We define the *DTB loss* $\mathcal{L}_{\nabla}(\mathfrak{g}_{\nabla}; \tau, \alpha)$ of the exploration GFlowNet $\mathfrak{g}_{\nabla}$ for a trajectory $\tau$ and threshold $\alpha > 0$ as

$$\left( \log \left( \frac{Z_{\phi}^{\nabla} \, p_F^{\nabla}(\tau; \phi)}{R(x)^{\beta} \, p_B^{\nabla}(\tau; \phi)} + \mathbb{I}[\tau \in \mathrm{OA}(\alpha)] \right) \right)^2. \quad (9)$$

We also define $\mathcal{L}_{\nabla}(\mathfrak{g}_{\nabla}; \mathfrak{g}, \alpha) = \mathbb{E}_{\tau \sim p_F^{\epsilon, \nabla}}[\mathcal{L}_{\nabla}(\mathfrak{g}_{\nabla}; \tau, \alpha)]$ as the average of $\mathcal{L}_{\nabla}$ with respect to the $\epsilon$-greedy version $p_F^{\epsilon, \nabla}$ of $p_F^{\nabla}$, wherein we make the dependence of $\mathcal{L}_{\nabla}$ on the canonical GFlowNet $\mathfrak{g}$ (via the set OA) explicit.

When optimized to zero, $\mathcal{L}_{\nabla}$ drives the exploratory policy $p_F^{\nabla}$ to sample proportionally to the reward in undersampled areas (i.e., $\mathrm{UA}(\alpha)$). We formalize this in Proposition 3.4.

**Proposition 3.4** (Complementary Sampling Property). *Assume $\mathcal{L}_{\nabla}(\mathfrak{g}_{\nabla}; \tau, \alpha) = 0$ for each trajectory $\tau$ starting at $s_o$ and finishing at $\mathcal{X}$ and $\mathrm{UA}(\alpha) \neq \emptyset$. Then, the marginal $p_{\top}^{\nabla}$ of $p_F^{\nabla}$ over $\mathcal{X}$ is*

$$p_{\top}^{\nabla}(x) \propto R(x)^{\beta} \cdot \mathbb{I}[x \in \mathrm{UA}(\alpha)],$$

*with normalizing constant $Z_{\nabla} = \sum_{x \in \mathrm{UA}(\alpha)} R(x)^{\beta}$.*

Interestingly, $p_{\top}^{\nabla}$'s serendipitous tendency described in Proposition 3.4—searching for high-reward states in unexplored regions—remains valid when the loss function is only approximately minimized. Theorem 3.5 shows this.

**Algorithm 1** Adaptive Complementary Exploration (ACE)

**Require:** Reward function $R(x)$, threshold $\alpha$
1: GFlowNets $\mathfrak{g} \leftarrow (\theta, Z_\theta)$ and $\mathfrak{g}_\nabla \leftarrow (\phi, Z_\phi^\nabla)$
2: **while** not converged **do**
3:     // Phase 1: Sampling
4:     $\mathcal{B} \leftarrow \{\tau \sim p_F(\cdot; \theta)\}$
5:     $\mathcal{B}_\nabla \leftarrow \{\tau \sim p_F^{\epsilon,\nabla}(\cdot; \phi)\}$
6:     // Phase 2: Exploitation Update
7:     Calculate mixing weight: $w \leftarrow \text{sg}\left(\frac{Z_\theta}{Z_\theta + Z_\phi}\right)$
8:     $\mathcal{L}_1 \leftarrow \frac{1}{|\mathcal{B}|} \sum_{\tau \in \mathcal{B}} \mathcal{L}_{\text{TB}}(\theta; \tau)$
9:     $\mathcal{L}_2 \leftarrow \frac{1}{|\mathcal{B}_\nabla|} \sum_{\tau \in \mathcal{B}_\nabla} \mathcal{L}_{\text{TB}}(\theta; \tau)$
10:    $\mathcal{L} \leftarrow w \cdot \mathcal{L}_1 + (1-w)\mathcal{L}_2$
11:    Update $\theta$ by a gradient step on $\nabla_\theta \mathcal{L}$.
12:    // Phase 3: Exploration Update
13:    $\mathcal{L}_{\text{exp}} \leftarrow \frac{1}{|\mathcal{B}_\nabla|} \sum_{\tau \in \mathcal{B}_\nabla} \mathcal{L}_\nabla(\phi; \tau, \alpha)$
14:    Update $\phi$ by a gradient step on $\nabla_\phi \mathcal{L}_{\text{exp}}$.
15: **end while**

**Proposition 3.5** (Approximate Complementary Sampling).
*Let $\epsilon > 0$. Assume $\mathcal{L}_\nabla(\mathfrak{g}_\nabla, \tau, \alpha) \leq \epsilon$ for every trajectory $\tau$, and define $\pi_\nabla(x) = {R(x)^\beta}/{Z_\nabla}$. Also, let $\kappa(\epsilon) = \exp\sqrt{\epsilon}$. Then, for every $x \in \text{UA}(\alpha)$,*

$$\pi_\nabla(x)\kappa(\epsilon)^{-1} \leq p_\top^\nabla(x) \leq \pi_\nabla(x)\kappa(\epsilon)$$

*and, for every $x \in \text{OA}(\alpha)$,*

$$0 \leq p_\top^\nabla(x) \leq \pi_\nabla(x)(\kappa(\epsilon) - 1).$$

*In particular, $\pi_\nabla(x)(1-o(\epsilon)) \leq p_\top^\nabla(x) \leq \pi_\nabla(x)(1+o(\epsilon))$ for $x \in \text{UA}(\alpha)$ and $p_\top^\nabla(x) \leq \pi_\nabla(x)o(\epsilon)$ otherwise. Notably, when $\epsilon = 0$, we recover the identity in Proposition 3.4.*

Additionally, from an information-theoretic perspective, minimizing the expectation of $\mathcal{L}_\nabla$ under a measure $\mu$ supported on trajectories in $\text{OA}(\alpha)$ may be interpreted as maximizing a Kullback-Leibler divergence (Kullback & Leibler, 1951) based on $\mu$.

**Proposition 3.6** (Repulsive Bound). *Let $\mu$ be a probability measure over trajectories supported on $\text{OA}(\alpha)$, and define $p_B^\nabla(\tau) = \pi(x)p_B^\nabla(\tau|x)$ as the backward trajectory probability, with $\pi(x) \propto R(x)$ as the normalized target. Then,*

$$\mathbb{E}_{\tau \sim \mu}\left[\left(\log\frac{p_F^\nabla(\tau)}{p_B^\nabla(\tau)} + \mathbb{I}[\tau \in \text{OA}(\alpha)]\right)^2\right] \tag{10}$$
$$\geq \left(\log(2) + \mathcal{D}_{KL}[\mu\|p_B^\nabla] - \mathcal{D}_{KL}[\mu\|p_M^\nabla]\right)^2,$$

*in which $p_M^\nabla = \frac{1}{2}(p_F^\nabla + p_B^\nabla)$ is an uniform mixture of $p_F^\nabla$ and $p_B^\nabla$. This quantity is minimized when the marginal distribution $p_\top^\nabla$ of $p_F^\nabla$ on $\mathcal{X}$ vanishes on $\text{OA}(\alpha)$.*

**Adaptive Complementary Exploration.** As mentioned earlier, we train the canonical GFlowNet $\mathfrak{g}$ on samples from

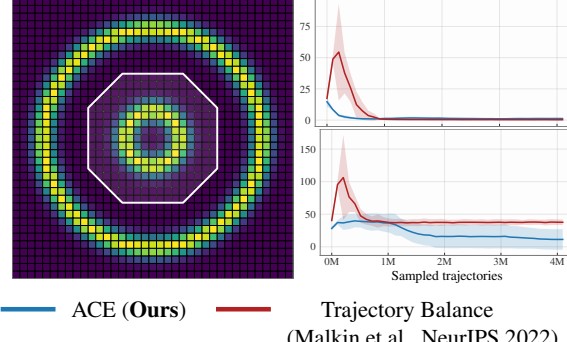

ACE (**Ours**)      Trajectory Balance
    (Malkin et al., NeurIPS 2022)

*Figure 2.* A GFlowNet trained on the RINGS distribution (left) via $\epsilon$-greedy exploration may overdraw samples from a well-approximated region (polygon), misrepresenting other high-probability regions. The TB residual on the rightmost panel for the inner (top) and outer (bottom) rings shows ACE avoids this issue.

both $\mathfrak{g}$ and $\mathfrak{g}_\nabla$ by generating trajectories from the mixture

$$p_F^{\text{ACE}}(s_o, \cdot) = w \cdot p_F(s_o, \cdot) + (1-w) \cdot p_F^{\epsilon,\nabla}(s_o, \cdot). \tag{11}$$

This raises the question: how to choose $w$ to better emphasize diverse high-reward states during training? Intuitively, we would like $w \to 1$ as the canonical GFlowNet $\mathfrak{g}$ covers a progressively large portion of the high-probability regions in the state space, and that $w < 0.5$ when $\mathfrak{g}_\nabla$ concentrates most of the probability mass in $\mathcal{X}$. By construction, under the light of Bengio et al. (2023, Proposition 10) and Proposition 3.4, we notice that the learned $Z$ and $Z_\nabla$ serve as proxies for the reward masses under $\mathfrak{g}$ and $\mathfrak{g}_\nabla$, respectively. With this in mind, we define $w$ as the *relative mass* under $\mathfrak{g}$,

$$w = \frac{Z}{Z_\nabla + Z}.$$

As we will show in Proposition 3.10, such a choice satisfies the desiderata above. Based on this, we define the loss function for the canonical GFlowNet below.

**Definition 3.7** (Canonical Loss). Let $\mathfrak{g}$ and $\mathfrak{g}_\nabla$ be the canonical and exploration GFlowNets. We define the *canonical loss* as the trajectory balance loss averaged over the mixture distribution $p_F^{\text{ACE}}$ in Equation (11), i.e.,

$$\mathcal{L}_{\text{CAN}}(\mathfrak{g}; \mathfrak{g}_\nabla) = \text{sg}(w) \cdot \mathbb{E}_{\tau \sim p_F}[\mathcal{L}_{\text{TB}}(\mathfrak{g}; \tau, R)]$$
$$+ \text{sg}(1-w) \cdot \mathbb{E}_{\tau \sim p_F^{\epsilon,\nabla}}[\mathcal{L}_{\text{TB}}(\mathfrak{g}; \tau, R)], \tag{12}$$

with sg as the stop-gradient operation, e.g., `jax.stop_gradient` in JAX (Bradbury et al., 2018) or `torch.Tensor.detach` in PyTorch (Paszke et al., 2019), which detachs $w$ from the computation graph.

We call *Adaptive Complementary Exploration* (ACE) the algorithm that learns both $\mathfrak{g}$ and $\mathfrak{g}_\nabla$ by minimizing the Canonical (Definition 3.7) and DTB (Definition 3.3) losses via Monte Carlo estimators based on their respective integrating measures. We summarize ACE in Algorithm 1.

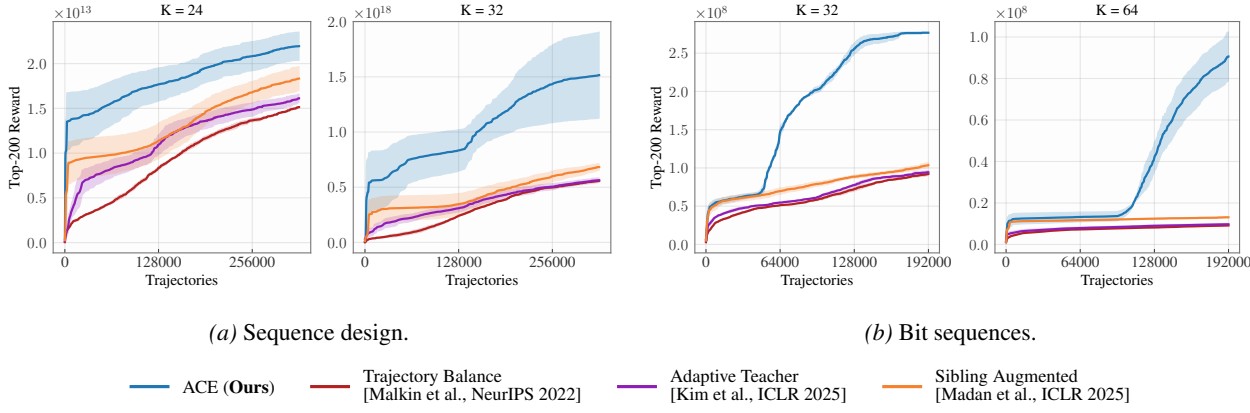

*(a)* Sequence design.                    *(b)* Bit sequences.

——— ACE (**Ours**)     Trajectory Balance     Adaptive Teacher     Sibling Augmented
                       [Malkin et al., NeurIPS 2022]     [Kim et al., ICLR 2025]     [Madan et al., ICLR 2025]

*Figure 3.* **ACE significantly accelerates mode-discovery for autoregressive sequence generation** with GFlowNets. Each plot shows the average reward of the unique 200 highest-valued discovered states as a function of the number of trajectories sampled throughout training.

There, to determine whether $\tau \in \mathrm{OA}(\alpha)$ in Equation (9), we use a single sample from $p_B(\cdot|x)$ to estimate $\hat{R}_{\mathfrak{g}}(x)$.

*Remark* 3.8 (Notation for the loss functions). To emphasize the parameterization of our models, we also denote by $\mathcal{L}_{\mathrm{TB}}(\theta; \tau)$ and $\mathcal{L}_{\nabla}(\phi; \tau, \alpha)$ the losses evaluated at $\tau$ for GFlowNets $\mathfrak{g}$ and $\mathfrak{g}_{\nabla}$ with parameters $\theta$ and $\phi$, respectively.

Notably, when $\mathfrak{g}$ and $\mathfrak{g}_{\nabla}$ collapse into a subset of $\mathcal{X}$, the expected on-policy gradient of $\mathcal{L}_{\nabla}$ pushes $\mathfrak{g}_{\nabla}$ towards the complement of that subset; see Proposition 3.9. Recall that we update $\phi$ via gradient steps in the direction of $-\nabla_{\phi}\mathcal{L}_{\nabla}$.

**Proposition 3.9.** *Assume $\mathfrak{g}$ and $\mathfrak{g}_{\nabla}$ are collapsed on a set $\mathrm{C} \subset \mathcal{X}$, i.e., $p_{\top}(\mathrm{C}) = 1$ and $p_{\top}^{\nabla}(\mathrm{C}) = 1$, that they satisfy their respective TB conditions on all trajectories leading to $\mathrm{C}$, and $\alpha < 1$. Then, if $\mathfrak{g}_{\nabla}$ is parameterized by $\phi = (\phi_F, \phi_B, Z_{\nabla})$, $\phi_F$ and $\phi_B$ as the parameters for $p_F^{\nabla}$ and $p_B^{\nabla}$,*

$$\mathbb{E}_{\tau \sim p_F^{\nabla}(\cdot; \phi_F)}\left[\nabla_{\phi_F}\mathcal{L}_{\nabla}(\phi; \tau, \alpha)\right] = -\log(2)\nabla_{\phi_F}p_{\top}^{\nabla}(\mathrm{C}^c; \phi_F),$$

*in which $p_{\top}^{\nabla}$ is as in Equation (1) and $\mathrm{C}^c := \mathcal{X} \setminus \mathrm{C}$.*

We also demonstrate that the equilibrium of the cooperative game implemented by Algorithm 1 depends on the choice of $\alpha$. If $\alpha \leq 1$, for instance, the repulsive force described in Proposition 3.6 forces $\mathfrak{g}_{\nabla}$ to collapse into $Z_{\nabla} = 0$. Otherwise, if $\alpha > 1$, the exploration GFlowNet matches the tempered target $R(x)^{\beta}$ on $\mathcal{X}$.

**Proposition 3.10** (Equilibrium State). *Assume $\mathfrak{g}^{\star} = (Z^{\star}, p_F^{\star}, p_B^{\star})$ and $\mathfrak{g}_{\nabla}^{\star} = (Z_{\nabla}^{\star}, p_F^{\nabla,\star}, p_B^{\nabla,\star})$ jointly satisfy*

$$\mathfrak{g}^{\star} = \arg\min_{\mathfrak{g}} \mathcal{L}_{\mathrm{CAN}}(\mathfrak{g}; \mathfrak{g}_{\nabla}^{\star}) \text{ and}$$

$$\mathfrak{g}_{\nabla}^{\star} = \arg\min_{\mathfrak{g}_{\nabla}} \mathcal{L}_{\nabla}(\mathfrak{g}_{\nabla}; \mathfrak{g}^{\star}, \alpha).$$

*Then, $Z^{\star} := \sum_{x \in \mathcal{X}} R(x)$ and $p_{\top}^{\star}(x) \propto R(x)$. When $\alpha \leq 1$, $Z_{\nabla}^{\star} = 0$. When $\alpha > 1$, $Z_{\nabla}^{\star} = \sum_{x \in \mathcal{X}} R(x)^{\beta}$ and $p_{\top}^{\nabla,\star}(x) \propto R(x)^{\beta}$, in which $p_{\top}^{\nabla,\star}$ is the marginal distribution over $\mathcal{X}$ induced by $\mathfrak{g}_{\nabla}^{\star}$ (recall Equation 1).*

Together, these results establish ACE as a principled approach for enhanced exploration of reward-dense regions during GFlowNet training. Importantly, the next section shows that ACE also consistently outperforms prior art on standard metrics used in the GFlowNet literature.

# 4. Experiments

We present a comprehensive empirical analysis of our method in this section. The central research questions (RQs) our campaign aims to respond are the following.

RQ1 Does ACE significantly speed up the number of diverse and high-reward states found during learning?

RQ2 Does ACE accelerate learning convergence?

We answer both in the affirmative by measuring the top-$K$ average reward of unique states found throughout training (Pan et al., 2023; Madan et al., 2022; 2025), the convergence rate of the log-partition function of the canonical GFlowNet, and the total variation (TV) distance between the learned and target distributions, defined as

$$\mathrm{TV}(p_{\top}, \pi) := \frac{1}{2}\sum_{x \in \mathcal{X}}|p_{\top}(x) - \pi(x)|,$$

in which $\pi(x) \propto R(x)$ is the normalized target and $p_{\top}$ is the GFlowNet's marginal over terminal states; see Equation (1).

Collectively, our experiments confirm that ACE is an effective algorithm that drastically improves the sample efficiency of GFlowNets. We refer the reader to Section C in the supplement for further details on our experiments.

## 4.1. Assessing ACE against GFlowNet variants

**Lazy Random Walk.** The state space $\mathcal{S}$ is $[[-m, m]]^d \times \{1, \dots, T-1\}$ for $m, d, T \in \mathbb{N}$ with $md \leq T$, and $\mathcal{X} := [[-m, m]]^d \times \{T\}$, and $[[-m, m]] =$

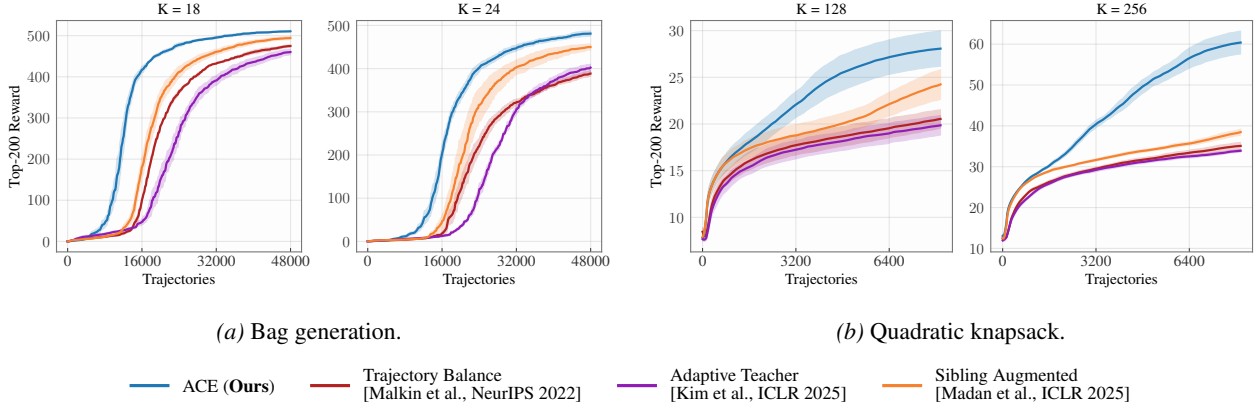

*(a)* Bag generation.  *(b)* Quadratic knapsack.

ACE (**Ours**) — Trajectory Balance [Malkin et al., NeurIPS 2022] — Adaptive Teacher [Kim et al., ICLR 2025] — Sibling Augmented [Madan et al., ICLR 2025]

*Figure 4.* **ACE finds diverse and high-reward states faster** than prior approaches for improved GFlowNet exploration for the bag generation (left) and quadratic knapsack (right) problems. In both (a) and (b), $K$ denotes the number of available items for selection.

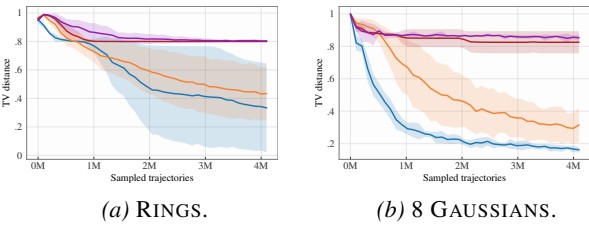

*(a)* RINGS.  *(b)* 8 GAUSSIANS.

*Figure 5.* **ACE results in faster learning convergence** than prior art for GFlowNet exploration for the Lazy Random Walk task. Please consult Figure 6 below for the legend.

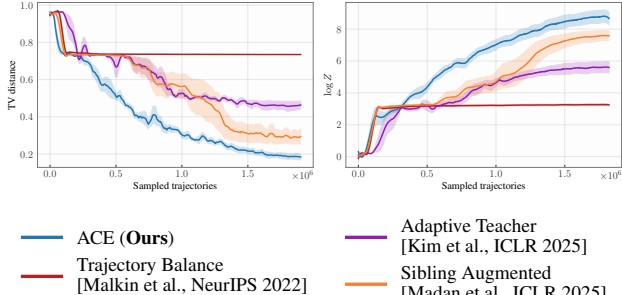

ACE (**Ours**) — Trajectory Balance [Malkin et al., NeurIPS 2022] — Adaptive Teacher [Kim et al., ICLR 2025] — Sibling Augmented [Madan et al., ICLR 2025]

*Figure 6.* **ACE achieves the best goodness-of-fit** to the target distribution in the grid world task described in Equation (13).

$\{-m, -m+1, \ldots, m\}$. The initial state is $s_o = (\mathbf{0}_d, 1) = ([0, \ldots, 0], 1)$ and each transition at $(\mathbf{s}, t)$ corresponds to either adding 1 or $-1$ to a chosen coordinate of $\mathbf{s}$ or staying in place; in either case, the counter $t$ is incremented to $t + 1$. This is repeated until $t = T$ (full details in Section C). We let $\alpha = 0.2$. In particular, we assess ACE on both RINGS—shown in Figure 2—and 8 GAUSSIANS distributions; Figure 5 highlights ACE achieves the best distributional fit.

**Grid world.** (Malkin et al., 2022; Madan et al., 2025) To further gauge ACE, we consider the standard grid world environment. There, $\mathcal{S} = [[0, H]]^d$ and $\mathcal{X} = \mathcal{S} \times \{\top\}$, in which $\top$ is an indicator of finality. The sampler starts at $s_o = \mathbf{0}$, and at each state $s$ we either add 1 to a coordinate of $s$ or transition to $x := s \times \{\top\} \in \mathcal{X}$. We let $H = 16$, $d = 2$, $y(x) = |5 \cdot x/H - 10|$, and train a GFlowNet to sample from

$$R(x) = 10^{-3} + 3 \cdot \prod_{1 \le i \le d} [y(x_i) \in (6, 8)] \quad (13)$$

in which $[C]$ represents Iverson's bracket, which evaluates to 1 if the clause $C$ is true and 0 otherwise; see Figure 1. Differently from Lazy Random Walk, the stopping action poses additional exploration challenges, as a randomly initialized sampler is less likely to encounter the distant (in Euclidean norm) modes from $s_o$ (Shen et al., 2023). Notably, Figure 6 shows that ACE results in faster training

convergence than both AT, SA, and $\epsilon$-greedy GFlowNets.

**Bit sequences.** (Malkin et al., 2022; Madan et al., 2022) We define $\mathcal{S} = \bigcup_{k \le K-1} \{1, 0\}^k$ and $\mathcal{X} = \{1, 0\}^K$ for a given sequence size $K$. As in Malkin et al. (2022), we let $\mathcal{M} \subseteq \mathcal{X}$ be a set of modes and $\log R(x) = \frac{1}{T} (1 - \min_{m \in \mathcal{M}} d(x, m)/K)$, in which $d(x, m)$ represents Levenshtein's distance between binary strings $x$ and $m$ and $T = 1/20$. Concretely, $\mathcal{S}$ represents the space of bit sequences with size up to $K - 1$. Starting at $s_o = []$, we append either 1 or 0 to the current state until it reaches the size of $K$. We consider $K \in \{32, 64\}$. Notably, Figure 3b shows that our method finds diverse high-reward states significantly faster than baselines.

**Sequence design.** (Silva et al., 2025) Similarly, $\mathcal{S} = \bigcup_{k \le K-1} \mathcal{V}^k$ for a finite vocabulary $\mathcal{V}$ and $K$, and $\mathcal{X} = \mathcal{V}^K$ with size $V := |\mathcal{V}|$. We consider $(K, V) \in \{(24, 6), (32, 4)\}$, and define the reward function of a $x \in \mathcal{X}$ through $\log R(x) := \sum_{k=1}^{K} u(k) \cdot v(x_k)$, in which $u: [K] \to \mathbb{R}$ and $v: \mathcal{V} \to \mathbb{R}$ are utility functions picked at random prior to training. (Recall $[K] = \{1, \ldots, K\}$). Remarkably, Figure 3a confirms that ACE significantly increase the discovery rate of high-reward regions for this task.

**Bag generation.** (Shen et al., 2023; Jang et al., 2024) A bag is a multiset with elements taken from a set $\mathcal{V}$. In Figure 4a, we let $\mathcal{S} := \{B \subseteq_{\text{multi}} \mathcal{V} : |\mathcal{B}| < S\}$ and $\mathcal{X} := \{B \subseteq_{\text{multi}} \mathcal{V} : |\mathcal{B}| = S\}$ be the set of $S$-sized multi-subsets of a set $\mathcal{V}$ with size $K$, and define $R(x) = \sum_{e \in x} u(e)$ for an utility function $u : \mathcal{V} \to \mathbb{R}_+$ drawn from a geometric Gaussian process. Once again, ACE improves upon prior approaches in terms of the speed with which high-probability regions are discovered.

**Quadratic Knapsack.** We present results for the Quadratic Knapsack task, which is an NP-hard problem (Caprara et al., 1999). Briefly, we let $W$ be the maximum weight, $\mathbf{w} \in \mathbb{R}_+^K$ be the weight of each of the $K$ items, $\mathbf{u} \in \mathbb{R}_+^K$ be their utilities, and $\mathbf{A} \in \mathbb{R}^{K \times K}$ be a symmetric matrix measuring the substitutability or complementarity of each item pair. We may choose up to $L$ copies of each item, and our objective is to find the item multiplicities $\mathbf{m} = (m_1, \ldots, m_K)$ maximizing the collection's utility, i.e.,

$$\max_{\mathbf{m} \in [[0,L]]^K} \langle \mathbf{u}, \mathbf{m} \rangle + \mathbf{m}^\top A \mathbf{m} \text{ s.t. } \langle \mathbf{m}, \mathbf{w} \rangle \leq W. \quad (14)$$

We let $K \in \{128, 256\}$ and $W = 60$. Our generative process starts at $s_o = \mathbf{0} \in \mathbb{R}^K$, and at each step we add an item to the current state $s$ until no items can be added (either due to repetition or weight limit). In this setting, $\mathcal{S} = \{\mathbf{m} \in [[0,L]]^K : \exists k, \langle \mathbf{m}, \mathbf{w} \rangle + \mathbf{w}_k \leq W \text{ and } \mathbf{m}_k + 1 \leq L\}$, and $\mathcal{X}$ is defined as the set for which such a $k$ does not exist, i.e., for which no more items can be added to the collection. The reward function $R$ being defined as the objective function in Equation (14). As previously noted, ACE exhibits the best sample efficiency in the search for high-valued feasible solutions for the Quadratic Knapsack problem.

**Antimicrobial Peptides (AMPs).** (Trabucco et al., 2022; Jain et al., 2022) We also evaluate ACE on the task of *de novo* design of AMPs potentially active against the following pathogens: *E. coli*, *S. aureus*, *P. aeruginosa*, *B. subtilis*, and *C. albicans*. The peptide design space is restricted to sequences with up to 10 amino acids. Given the standard proteinogenomic alphabet, consisting of 20 standard amino acids, this results in a search space with $10^{13}$ candidates.

Formally, the initial state is $s_o = [\,]$, the state space $\mathcal{S}$ consists of all amino acid sequences of size up to $L = 10$, and $\mathcal{X} = \{s \oplus \langle \text{EOS} \rangle : s \in \mathcal{S}\}$, in which $\langle \text{EOS} \rangle$ is a special end-of-sequence token and $\oplus$ represents the concatenation operator; for ACE, we let $\alpha = 0.2$ and $\beta = 1$. The reward is derived from a Random Forest classifier (Pedregosa et al., 2011; Dall'Antonia et al., 2025) predicting antimicrobial activity across pathogens (full details in Section C); we use a cutoff $c = 0.95$ and call a sequence a *mode* if its predicted activity probability satisfies $p(s) \geq c$.

Crucially, Figure 8 shows that ACE discovers substantially more unique high-reward AMPs throughout training

than all competing methods. On the log-scaled $y$-axis, ACE achieves an order-of-magnitude improvement in the cumulative number of unique effective peptides, indicating both faster mode discovery and sustained exploration. To further assess diversity, Figure 7 visualizes the modes found during training via a 2D UMAP projection of their $k$-mer frequency embeddings, highlighting the significantly broader coverage achieved by ACE.

### 4.2. Understanding ACE: Ablations, Stability, and Hyperparameter Selection

To conclude, we address practical questions regarding ACE's training behavior and empirically study the influence of $\alpha$, $\beta$, and the number of backward trajectories used for the estimation of $\hat{R}_{\mathfrak{g}}$ on learning stability.

**How do** OA **and** UA **evolve during training?** To further understand the character of $\text{OA}(\alpha)$ (and $\text{UA}(\alpha) := \mathcal{X} \setminus \text{OA}(\alpha)$) as a function of $\alpha$, we depict in Figure 9 the evolution of $\text{OA}(\alpha)$ during learning. As expected from Proposition 3.10, when $\alpha < 1$, $\text{OA}(\alpha) \to \mathcal{X}$ and, when $\alpha > 1$, $\text{OA}(\alpha) \to \emptyset$. Intuitively, the reason for this is that $Z \cdot p_F(\tau) \to R(x) \cdot p_B(\tau|x)$ for each $\tau, x$ as training progresses; hence, if $\alpha < 1$ (resp. $\alpha > 1$), each state will be eventually over-allocated (resp. under-allocated).

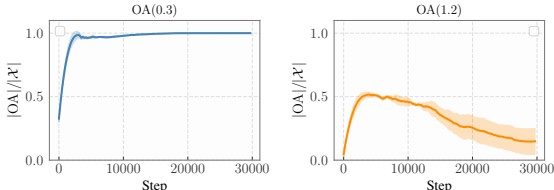

*Figure 9.* $|\text{OA}(\alpha)|/|\mathcal{X}|$ throughout training for the $16 \times 16$ grid world.

**How does our estimate of** $\hat{R}_{\mathfrak{g}}$ **affect learning?** As mentioned, exact computation of $\hat{R}_{\mathfrak{g}}(x)$ is intractable, and Algorithm 1 relies on a noisy membership test to decide whether $x \in \text{OA}(\alpha)$ based on trajectories $\tau$ drawn from $p_B(\tau|x)$. Although we could enhance this test's accuracy by increasing the number of sampled $\tau$'s, we have found that the extra computational cost does not lead to a corresponding improvement in learning convergence; see Figure 10. Our recommendation is thus to use a single trajectory for estimating $\hat{R}_{\mathfrak{g}}(x)$, which we observed to be effective.

**How to select** $\alpha$ **and** $\beta$**?** ACE introduces two additional hyperparameters: $\alpha$ and $\beta$. We also provide a comprehensive ablation study for $\alpha$ and $\beta$ in Section D. In practice, we found $\alpha = 0.3$ and $\beta = 0.25$ to be effective in most experiments, and recommend this as the default; a small $\beta$ is particularly suggested for sparse, multi-modal distributions, which are often encountered in applications. Importantly, ACE has fewer hyperparameters than both AT- and SA-GFlowNets, our primary baselines.

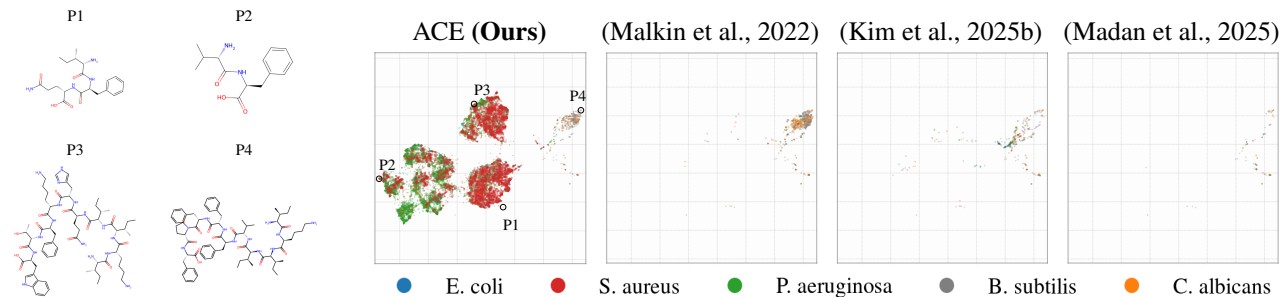

Figure 7. **Peptide embeddings colored by predicted micro-organism group.** Each dot represents the 2D projection of the k-mer embedding of the AMPs in Figure 8. As we can see, ACE generates a substantially more diverse set of AMPs with likely antimicrobial activity than alternative methods. We showcase a subset of ACE's generated peptides in in the leftmost panel.

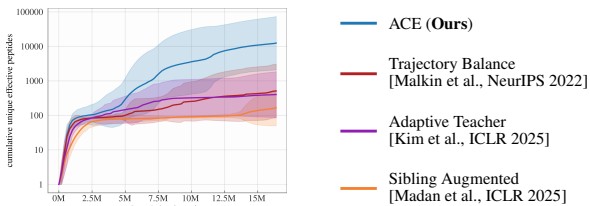

Figure 8. **ACE identifies a larger number of high-fitness AMPs** than prior methods. The plot shows the number of unique AMPs found during training with at least 95% probability of exhibiting antimicrobial activity as a function of the number of trajectories.

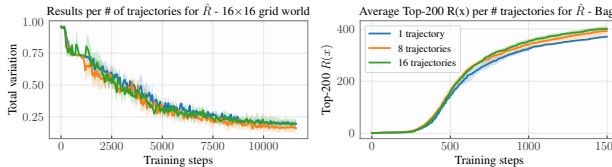

Figure 10. TV & Top-200 $R(x)$ for the grid world ($H = 16$) and bags ($K = 24$), respectively, as a function of the number of trajectories for estimating $\hat{R}_\mathfrak{g}(x)$ in Algorithm 1.

We also show how to apply ACE to continuous domains in Section E in the supplement.

## 5. Discussion

We introduced *Adaptive Complementary Exploration* (ACE) as a principled algorithm for effective exploration of under-explored regions during the training of GFlowNets. While former approaches focused on learning an exploratory policy via curiosity-driven methods, which we have shown may overemphasize well-approximated regions of the state space (e.g., Figure 1), ACE promotes the visitation of novel states through the newly proposed *divergent trajectory balance* (DTB) loss. Importantly, we proved that the minimization of DTB pushes the exploratory policy away from oversampled trajectories by the canonical GFlowNet, providing a rigorous foundation for our method.

Our experiments demonstrated ACE consistently and significantly outperformed prior approaches for improved GFlowNet exploration in terms of the discovery rate of diverse, high-reward regions and the goodness-of-fit to the target distribution. In conclusion, we also believe that exploring whether a non-stationary reward (e.g., curiosity-driven) can be used in Definition 3.2 and how to optimally weight the GFlowNets' samples in Equation (11) are promising directions for future research.

## Impact Statement

ACE significanlty improves the sample efficiency of GFlowNet training. As GFlowNets are increasingly applied to scientific discovery—where combinatorial structures such as graphs and sequences are widespread but reward evaluation is often costly—we believe our method could yield substantial computational benefits. That said, we do not foresee any immediate ethical or societal concerns.

## Acknowledgements

PD and DM acknowledge the support by the Fundação Carlos Chagas Filho de Amparo à Pesquisa do Estado do Rio de Janeiro (FAPERJ) (SEI-260003/020348/2025, SEI-260003/020694/2025) and the Conselho Nacional de Desenvolvimento Científico e Tecnológico (CNPq) (404336/2023-0, 305692/2025-9).

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

# A. Proofs

We provide rigorous proofs for each of our statements in this section.

## A.1. Proof of Proposition 3.4

We will demonstrate that $\mathcal{L}_\nabla(\mathfrak{g}_\nabla; \tau, \alpha) = 0$ for every $\tau$ if and only if $p_F^\nabla$ is supported on trajectories $\tau$ resulting in the under-allocated set $\mathrm{UA}(\alpha, \mathfrak{g})$, and that $p_\top^\nabla(x) \propto R(x)^\beta$ for $x$ in such a set.

To see this, we rewrite $\mathcal{L}_\nabla$ as

$$\lambda \mathbb{E}_{\tau \sim p_F^{\epsilon,\nabla}}\left[\left(\log \frac{Z_\nabla p_F^\nabla(\tau)}{R(x)p_B^\nabla(\tau|x)}\right)^2 \middle| \tau \in \mathrm{UA}(\alpha)\right] + (1-\lambda)\mathbb{E}_{\tau \sim p_F^{\epsilon,\nabla}}\left[\left(\log\left(\frac{Z_\nabla p_F^\nabla(\tau)}{R(x)p_B^\nabla(\tau|x)} + 1\right)\right)^2 \middle| \tau \in \mathrm{OA}(\alpha)\right],$$

in which $\lambda := p_\top^{\epsilon,\nabla}(\mathrm{UA}(\alpha)) \geq \epsilon p_\top^U(\mathrm{UA}(\alpha)) > 0$, with $p_\top^U$ denoting the marginal distribution over $\mathcal{X}$ induced by the uniform policy. The first member of the above equation is globally minimized when $Z_\nabla p_F^\nabla(\tau) = R(x)p_B^\nabla(\tau|x)$ for $\tau \in \mathrm{UA}(\alpha)$, which in particular yields $Z_\nabla \neq 0$ as $\mathrm{UA}(\alpha)$ is non-empty by assumption. When $Z_\nabla \neq 0$, the second member is minimized when $p_F^\nabla(\tau) = 0$ if $\tau \in \mathrm{UA}(\alpha)$. As $x \mapsto x^2$ is non-negative, this is the only global minimizer of the DTB loss.

Under these circumstances, the marginal distribution of $\mathfrak{g}_\nabla$ over $\mathcal{X}$ is

$$\mathbb{I}[x \in \mathrm{UA}(\alpha)] \sum_{\tau:\, s_o \rightsquigarrow x} p_F^\nabla(\tau) = \mathbb{I}[x \in \mathrm{UA}(\alpha)] \cdot \sum_{\tau:\, s_o \rightsquigarrow x} \frac{R(x)^\beta}{Z} p_B(\tau|x) \propto \mathbb{I}[x \in \mathrm{UA}(\alpha)] \cdot R(x)^\beta,$$

(by Malkin et al. (2022, Proposition 1)) with normalizing constant $Z_\nabla = \sum_{x \in \mathcal{X}} R(x)^\beta$. Importantly, the above computation is valid since, by Definition 3.1, $x \in \mathrm{OA}(\alpha)$ implies that $\tau \in \mathrm{OA}(\alpha)$ for each $\tau$ going from $s_o$ to $x$.

## A.2. Proof of Proposition 3.5

We will demonstrate that, when the exploratory network's loss function is uniformly upper-bounded by $\epsilon > 0$, the $p_\top^\nabla$ is approximately equal to the distribution $R(x)^\beta$ restricted to the set $\mathrm{UA}(\alpha)$ of under-allocated probability mass. For this, notice that

$$\left(\log\left(\frac{Z_\nabla p_F^\nabla(\tau)}{R(x)^\beta p_B^\nabla(\tau|x)} + \mathbb{I}[\tau \in \mathrm{OA}(\alpha)]\right)\right)^2 \leq \epsilon$$

for each $\tau, x$ implies

$$\exp\left\{-\sqrt{\epsilon}\right\} \leq \frac{Z_\nabla p_F^\nabla(\tau)}{R(x)^\beta p_B^\nabla(\tau|x)} + \mathbb{I}[\tau \in \mathrm{OA}(\alpha)] \leq \exp\sqrt{\epsilon}.$$

Define $\kappa(\epsilon) = \exp\sqrt{\epsilon}$. Then,

$$\kappa(\epsilon)^{-1} - \mathbb{I}[\tau \in \mathrm{OA}(\alpha)] \leq \frac{Z_\nabla p_F^\nabla(\tau)}{R(x)^\beta p_B^\nabla(\tau|x)} \leq \kappa(\epsilon) - \mathbb{I}[\tau \in \mathrm{OA}(\alpha)].$$

As $\kappa(\epsilon) \geq 1$, $\kappa(\epsilon)^{-1} \leq \mathbb{I}[\tau \in \mathrm{OA}(\alpha)]$ if $\tau \in \mathrm{OA}(\alpha)$, in which case the above inequality becomes

$$0 \leq \frac{Z_\nabla p_F^\nabla(\tau)}{R(x)^\beta p_B^\nabla(\tau|x)} \leq \kappa(\epsilon) - 1$$

for $\tau, x \in \mathrm{OA}(\alpha)$. Multiplying the members of this inequality by $R(x)^\beta p_B^\nabla(\tau|x)$ and summing over $\tau:\, s_o \rightsquigarrow x$, we infer

$$0 \leq p_\top^\nabla(x) \leq \pi_\nabla(x)(\kappa(\epsilon) - 1),$$

with $\pi_\nabla(x) = R(x)^\beta/Z_\nabla$, as defined in Proposition 3.5's statement. Similarly, for $\tau, x \in \mathrm{UA}(\alpha)$,

$$\kappa(\epsilon)^{-1} \leq \frac{Z_\nabla p_F^\nabla(\tau)}{R(x)p_B^\nabla(\tau|x)} \leq \kappa(\epsilon).$$

Again, multiplying all members by $R(x)p_B^\nabla(\tau|x)$ and summing over $\tau:\, s_o \rightsquigarrow x$, we obtain

$$\kappa(\epsilon)^{-1}\pi_\nabla(x) \leq p_\top^\nabla(x) \leq \kappa(\epsilon)\pi_\nabla(x).$$

To conclude, recall $\kappa(\epsilon) = 1 + o(\epsilon)$ and $\kappa(\epsilon)^{-1} = 1 - o(\epsilon)$ (Puiseux expansion of $x \mapsto \exp\sqrt{x}$). This shows Proposition 3.5.

### A.3. Proof of Proposition 3.6

We will first demonstrate that, for any measure $\mu$ supported on $\mathrm{OA}(\alpha)$,

$$\mathbb{E}_{\tau\sim\mu}\left[\left(\log\left(\frac{p_F(\tau)}{\pi(x)p_B(\tau|x)}+\mathbb{I}[\tau\in\mathrm{OA}(\alpha)]\right)\right)^2\right]\geq\left(\log(2)+\mathcal{D}_{\mathrm{KL}}\left[\mu||p_B\right]-\mathcal{D}_{\mathrm{KL}}[\mu||p_M]\right)^2, \tag{15}$$

in which $p_B(\tau)\coloneqq\pi(x)p_B(\tau|x)$ is the probability distribution over trajectories induced by $p_B$ and the target $\pi(x)\propto R(x)$ and $p_M(\tau)\coloneqq\mathbf{1}/2p_F(\tau)+\mathbf{1}/2p_B(\tau)$ is the arithmetic average between $p_F$ and $p_B$. Also, $\mathcal{D}_{\mathrm{KL}}$ is the standard Kullback-Leibler divergence, defined as

$$\mathcal{D}_{\mathrm{KL}}[p||q]=\mathbb{E}_{\tau\sim p}\left[\log\frac{p(\tau)}{q(\tau)}\right].$$

To understand Equation (15), notice that $\mathbb{I}[\tau\in\mathrm{OA}(\alpha)]=1$ $\mu$-almost surely and

$$\begin{aligned}\mathbb{E}_{\tau\sim\mu}\left[\log\left(\frac{p_F(\tau)}{p_B(\tau)}+1\right)\right]&=\mathbb{E}_{\tau\sim\mu}\left[\log\left(\frac{p_F(\tau)+p_B(\tau)}{p_B(\tau)}\right)\right]\\&=\mathbb{E}_{\tau\sim\mu}\left[\log\left(\frac{p_F(\tau)+p_B(\tau)}{p_B(\tau)}\right)+\log\frac{\mu(\tau)}{\mu(\tau)}\right]\\&=\mathbb{E}_{\tau\sim\mu}\left[\log\left(\frac{p_F(\tau)+p_B(\tau)}{\mu(\tau)}\right)\right]+\mathbb{E}_{\tau\sim\mu}\left[\log\left(\frac{\mu(\tau)}{p_B(\tau)}\right)\right]\\&=\log(2)-\mathbb{E}_{\tau\sim\mu}\left[\log\left(\frac{2\mu(\tau)}{p_F(\tau)+p_B(\tau)}\right)\right]+\mathbb{E}_{\tau\sim\mu}\left[\log\left(\frac{\mu(\tau)}{p_B(\tau)}\right)\right]\\&=\log(2)-\mathcal{D}_{\mathrm{KL}}[\mu||p_M]+\mathcal{D}_{\mathrm{KL}}[\mu||p_B].\end{aligned}$$

By Jensen's inequality and the nonnegativity of $x\mapsto\log(1+x)$ for $x\geq0$,

$$\begin{aligned}\mathbb{E}_{\tau\sim\mu}\left[\left(\log\left(\frac{p_F(\tau)}{p_B(\tau)}+1\right)\right)^2\right]&\geq\mathbb{E}_{\tau\sim\mu}\left[\log\left(\frac{p_F(\tau)}{p_B(\tau)}+1\right)\right]^2\\&=\left(\log(2)-\mathcal{D}_{\mathrm{KL}}[\mu||p_M]+\mathcal{D}_{\mathrm{KL}}[\mu||p_B]\right)^2.\end{aligned} \tag{16}$$

This proves our information-theoretic lower bound for the DTB loss. Clearly, when $p_F=0$ on the support of $\mu$,

$$\mathcal{D}_{\mathrm{KL}}[\mu||p_B]=\mathcal{D}_{\mathrm{KL}}[\mu||p_M]-\log(2),$$

which minimizes the right-hand side (RHS) of Equation (16). When $p_F(\tau)>0$ for a certain $\tau$ for which $\mu(\tau)>0$, $p_B(\tau)<p_F(\tau)+p_B(\tau)$ and $\log(2)+\mathcal{D}_{\mathrm{KL}}[\mu||p_B]>\mathcal{D}_{\mathrm{KL}}[\mu||p_M]$; hence, such a $p_F$ does not minimize $(\log(2)-\mathcal{D}_{\mathrm{KL}}[\mu||p_M]+\mathcal{D}_{\mathrm{KL}}[\mu||p_B])^2$. As a consequence, $p_F=0$ on the support $\mathrm{OA}(\alpha)$ of $\mu$ is the only minimizer of the RHS of Equation (16).

### A.4. Proof of Proposition 3.9

We will demonstrate that the on-policy expected gradient of the DTB loss function pushes the exploration GFlowNet $\mathfrak{g}_\nabla$ towards the complement of the canonical GFlowNet $\mathfrak{g}$'s support when $\mathfrak{g}$'s is collapsed into a subset C of $\mathcal{X}$. For this, first notice that, since $p_\top(\mathrm{C})=1$, $p_F(\tau)=0$ for each $\tau$ resulting in $\mathrm{C}^c\coloneqq\mathcal{X}\setminus\mathrm{C}$; otherwise, $p_\top(\mathrm{C}^c)\geq p_F(\tau)>0$ and $p_\top(\mathrm{C})=1-p_\top(\mathrm{C}^c)<1$.

As in Definition 3.1, we will adopt the convention that $\tau\in\mathrm{C}$ if $\tau$ leads up to a state $x\in\mathrm{C}$. As $p_F$ satisfies the TB condition for the trajectories in C, $Zp_F(\tau)=R(x)p_B(\tau|x)$ for $\tau\in\mathrm{C}$. By our previous observation, $p_F(\tau)=0$ for $\tau\in\mathrm{C}^c$. As a consequence, since $\alpha<1$, we have $\mathrm{UA}(\alpha,\mathfrak{g})=\mathrm{C}^c$ and $\mathrm{OA}(\alpha,\mathfrak{g})=\mathrm{C}$. As such, the loss function $\mathcal{L}_\nabla$ for $\mathfrak{g}_\nabla$ becomes

$$\mathcal{L}_\nabla(\phi;\tau,\alpha)=\begin{cases}\mathcal{L}_{\mathrm{TB}}(\phi;\tau)\text{ if }\tau\in\mathrm{C}^c,\\\mathcal{L}_{\mathrm{SP}}(\phi;\tau)=\mathrm{softplus}\left(\log\frac{Zp_F^\nabla(\tau)}{R(x)^\beta p_B^\nabla(\tau|x)}\right)^2\text{ otherwise,}\end{cases}$$

in which $\mathrm{softplus}$ is the mapping $x \mapsto \log(\exp\{x\} + 1)$. As $p_F^\nabla$ is collapsed into C, only the second term matters for our calculations. Also, if $Q(\phi; \tau) = \frac{Z_\nabla p_F^\nabla(\tau)}{R(x)p_B^\nabla(\tau|x)}$,

$$\nabla_{\phi_F} \mathcal{L}_{\mathrm{SP}}(\phi; \tau) = 2 \cdot \log(Q(\phi; \tau) + 1) \cdot \frac{1}{Q(\phi; \tau) + 1} \cdot \nabla_{\phi_F} Q(\phi; \tau).$$

Based on our assumptions, $Q(\phi; \tau) = 1$ for $\tau \in \mathrm{C}$; hence,

$$\nabla_{\phi_F} \mathcal{L}_{\mathrm{SP}}(\phi; \tau) = \log(2) \cdot \nabla_{\phi_F} Q(\phi; \tau)$$
$$= \log(2) \cdot \frac{Z}{R(x)p_B^\nabla(\tau|x)} \cdot \nabla_{\phi_F} p_F(\tau; \phi_F)$$
$$= \log(2) \cdot \frac{1}{p_F^\nabla(\tau; \phi_F)} \cdot \nabla_{\phi_F} p_F^\nabla(\tau; \phi_F).$$

In this scenario, the expectation of $\nabla_{\phi_F} \mathcal{L}_\nabla$ with respect to $p_F(\tau; \phi_F)$ is

$$\mathbb{E}_{\tau \sim p_F^\nabla(\cdot; \phi_F)} [\nabla_{\phi_F} \mathcal{L}_\nabla(\phi; \tau, \alpha)] = \mathbb{E}_{\tau \sim p_F^\nabla(\cdot; \phi_F)} [\nabla_{\phi_F} \mathcal{L}_{\mathrm{SB}}(\phi; \tau)]$$
$$= \mathbb{E}_{\tau \sim p_F^\nabla(\cdot; \phi_F)} \left[ \log(2) \cdot \frac{1}{p_F^\nabla(\tau; \phi_F)} \cdot \nabla_{\phi_F} p_F^\nabla(\tau; \phi_F) \right]$$
$$= \sum_{x \in \mathrm{C}} \sum_{\tau: \, s_o \rightsquigarrow x} p_F^\nabla(\tau; \phi_F) \cdot \log(2) \cdot \frac{1}{p_F^\nabla(\tau; \phi_F)} \cdot \nabla_{\phi_F} p_F^\nabla(\tau; \phi_F)$$
$$= \log(2) \sum_{x \in \mathrm{C}} \sum_{\tau: \, s_o \rightsquigarrow x} \nabla_{\phi_F} p_F^\nabla(\tau; \phi_F)$$
$$= \log(2) \cdot \nabla_{\phi_F} \sum_{x \in \mathrm{C}} p_\top^\nabla(x)$$
$$= \log(2) \cdot \nabla_{\phi_F} \left( 1 - \sum_{x \in \mathrm{C}^c} p_\top^\nabla(x; \phi_F) \right)$$
$$= -\log(2)\nabla_{\phi_F} \sum_{x \in \mathrm{C}^c} p_\top^\nabla(x; \phi_F) = -\log(2) \cdot \nabla_{\phi_F} p_\top^\nabla(\mathrm{C}^c).$$

This shows that, under Proposition 3.9 conditions, the on-policy expected gradient of the DTB loss for the exploration GFlowNet points in the direction of decreasing probability mass in $\mathrm{C}^c$. As we optimize $\phi$ via gradient descent on $\mathcal{L}_\nabla$, our algorithm moves in the direction of increasing the probability mass in $\mathrm{C}^c$ according to the exploration GFlowNet's model.

### A.5. Proof of Proposition 3.10

We fix $\mathfrak{g}_\nabla = (Z_\nabla, p_F^\nabla, p_B^\nabla)$. Then, the loss function

$$\mathcal{L}_{\mathrm{CAN}}(\mathfrak{g}; \mathfrak{g}_\nabla)$$

is minimized when $Z^\star = \sum_{x \in \mathcal{X}} R(x)$ and $Z^\star p_F^\star(\tau) = p_B^\star(\tau|x)R(x)$ for each complete trajectory $\tau: s_o \rightsquigarrow x$. Under these conditions, the set of trajectories with over-allocated mass,

$$\mathrm{OA}(\alpha, \mathfrak{g}^\star) = \{\tau: Z^\star p_F^\star(\tau) \geq \alpha R(x)p_B^\star(\tau|x)\}$$

either contains every trajectory (in case $\alpha \leq 1$) or none (if $\alpha > 1$). In the former case, the loss function for ACE reduces to

$$\mathbb{E}_{\tau \sim p_F^{\epsilon, \nabla}} \left[ \left( \log \left( \frac{Z_\nabla p_F^\nabla(\tau)}{R(x)^\beta p_B(\tau|x)} + 1 \right) \right)^2 \right],$$

which is minimized by $Z_\nabla^\star = 0$. In the latter case, the loss function for ACE becomes

$$\mathbb{E}_{\tau \sim p_F^{\epsilon, \nabla}} \left[ \left( \log \frac{Z_\nabla p_F^\nabla(\tau)}{R(x)^\beta p_B^\nabla(\tau|x)} \right)^2 \right],$$

which is the standard TB loss (Malkin et al., 2022) under an $\epsilon$-greedy policy, minimized when $Z_\triangledown^\star = \sum_{x \in \mathcal{X}} R(x)^\beta$ and the marginal of $p_F^{\triangledown,\star}(s_o, \cdot)$ over $\mathcal{X}$ satisfies $p_\top^{\triangledown,\star}(x) \propto R(x)^\beta$. Conversely, if our exploration GFlowNet satisfies either of these conditions, it should be clear by Malkin et al. (2022, Proposition 1) that the optimal canonical GFlowNet is the one satisfying $p_\top^\star(x) \propto R(x)$ and $Z^\star = \sum_{x \in \mathcal{X}} R(x)$. Indeed, we separate our demonstration into the following cases.

1. If $\alpha > 1$ and $p_\top^\triangledown \propto R(x)^\beta$, the best $p_F^\star$ minimizing the GFlowNet's canonical loss in Definition 3.7 satisfies $p_\top^\star(x) \propto R(x)$ and $Z = \sum_{x \in \mathcal{X}} R(x)$ since $R(x) > 0$ is a positive measure on $\mathcal{X}$.

2. Otherwise, if $Z_\triangledown^\star = 0$, then the weighting parameter $w = 1$ and the GFlowNet is trained via TB on-policy by Definition 3.7. By Proposition 3.4, $Z_\triangledown = 0$ is only optimal as long as the GFlowNet $\mathfrak{g}$ maintains full support over the space of trajectories. Under this condition, the only minimizer of the canonical loss is the GFlowNet $\mathfrak{g}$ satisfying $p_\top(x) \propto R(x)$ and $Z = \sum_{x \in \mathcal{X}} R(x)$.

As such, we have shown via a fixed-point-based argument that both of these are equilibria solutions to the minimization problem stated in Proposition 3.10.

## B. Related Works

Generative Flow Networks (GFlowNets; Bengio et al., 2021; 2023; Lahlou et al., 2023) have found successful applications in combinatorial optimization (Zhang et al., 2023a;b), causal discovery (Deleu et al., 2022; 2023), biological sequence design (Jain et al., 2022), and LLM finetuning (Hu et al., 2023; Venkatraman et al., 2024). They have also found promising applications in the AI for Science community (Jain et al., 2023; Wang et al., 2023), as illustrated in the task for *de novo* design of AMPs in Section 4, with their relationship to variational inference and reinforcement learning being formally established by Tiapkin et al. (2024); Malkin et al. (2023); Zimmermann et al. (2023). In this context, many studies focused on improving the sample efficiency of GFlowNets (e.g., (Pan et al., 2023; 2024; Hu et al., 2025; Madan et al., 2022; Zhang & Cao, 2025)), while others outlined their limitations (Kim et al., 2025b; Silva et al., 2025; Shen et al., 2023; Yu, 2025), with the effective exploration of diverse and high-reward states often being the main empirical concern. Although previous works have previously considered training multiple GFlowNets concomitantly to speed up learning (Lau et al., 2023; Madan et al., 2025; Kim et al., 2025c; Malek et al., 2026), ACE is—to the best of our knowledge—the first approach that directly promotes novelty through a penalty term that enforces a diversity-inducing balance condition, which we call Divergent Trajectory Balance.

## C. Experimental Details

This section provides further experimental details for our empirical analysis in Section 4.

### C.1. Optimization & Architecture

**Architecture.** Across all environments we parameterize the forward and backward policies $(p_F, p_B)$ with lightweight neural networks producing action logits.

For the *Lazy Random Walk* environment, both $p_F$ and $p_B$ use `FourierTimePolicy`. The policy input is obs $= (x, y, \tau)$, where $(x, y) \in \mathbb{R}^2$ are the current coordinates and $\tau \in [0, 1]$ is a (clipped) normalized time variable. We construct Fourier features with frequencies $f_k = 2^k$ for $k = 0, \ldots, n_{\text{freq}} - 1$:

$$\phi(\tau) = \left[ \mathbf{1}_{\text{include\_tau}} \tau, \ \{\sin(2\pi\tau f_k)\}_k, \ \{\cos(2\pi\tau f_k)\}_k \right],$$

concatenate them with $(x, y)$, and pass the result through an MLP with `num_layers` layers, `hidden_dim` hidden units, and ReLU activations, outputting logits over 5 discrete actions.

For *AMPs*, we use a windowed MLP policy. Given a padded sequence $s \in \{0, \ldots, V - 1\}^L$ with PAD/EOS $= 0$, we compute the current length $\ell = \sum_{t=1}^L \mathbf{1}[s_t \neq 0]$, embed the last $W = 6$ tokens with an embedding of dimension $D = 64$, flatten the resulting $W \times D$ representation, concatenate a sinusoidal positional encoding $\text{PE}(\ell) \in \mathbb{R}^{d_{\text{pos}}}$ with $d_{\text{pos}} = 16$, and map to logits over the vocabulary via a two-layer MLP $(WD + d_{\text{pos}}) \to 128 \to V$ with ReLU.

For *all other environments*, we follow the same setup as before: $p_F$ and $p_B$ are parameterized by an MLP with two hidden layers and 128 hidden units per layer, using Leaky ReLU activations throughout.

**Optimization** Across all environments, we optimize the GFlowNet policy parameters (i.e., those of $p_F$ and $p_B$) and the log-partition estimate $\log Z$ with separate optimizers, and we use AdamW (Loshchilov & Hutter, 2019) throughout.

For the *Lazy Random Walk* environment, we use learning rates $5 \times 10^{-3}$ for the policy parameters and $5 \times 10^{-2}$ for $\log Z$, together with a linear learning-rate schedule that decays from a factor of $1.0$ to $0.1$ over the training horizon applied to both optimizers.

For *AMPs*, we use fixed learning rates $0.05$ for the forward policy and $0.1$ for $\log Z$, without any scheduler.

For *all other environments*, we follow the same setup as before: AdamW with learning rate $10^{-2}$ linearly decayed to $10^{-4}$ for the policy parameters, and a learning rate $10\times$ larger for $\log Z$.

To ensure a fair comparison between methods, both ACE and SA GFlowNets are trained with a batch size equal to half that of AT and TB GFlowNets.

**Random seeds and uncertainty bands.** Unless otherwise stated, all curves report the mean over multiple random seeds and an uncertainty band corresponding to $\pm 1$ standard deviation across seeds. For the *AMP* experiments we use 15 seeds, from 10 to 24 For the *Lazy Random Walk* experiments we use seeds 5, $\{42, 43, 44, 45, 46\}$. For *all other environments* we use 3 seeds $\{42, 126, 210\}$. For the *AMP* plots reported on a log scale, we compute the mean and standard deviation in the log-domain, so that the displayed $\pm 1$ standard deviation band is symmetric in log space.

### C.2. Environment Specifications

The experimental setup for each environment was described in Section 4 in the main text. The number of training iterations was set to 5000 for sequence design, 3000 for bit sequences, 256 for knapsack, 30000 for grid world, 1500 for bags, 4000 for both lazy random walk and *AMPs*. Across all tasks, we use $\epsilon$-greedy exploration: we set $\epsilon = 0.3$ for *AMPs*, $\epsilon = 0.1$ for *Lazy Random Walk*, and $\epsilon = 0.05$ for all other environments. We use $\beta = 1$ on *AMPs* and $\beta = 0.25$ on all remaining environments. For $\alpha$, we set $\alpha = 0.2$ for *AMPs* and *Lazy Random Walk*, and $\alpha = 0.3$ for the rest.

**Random forest classifier for antimicrobial activity.** The proxy reward function is based on a Random Forest classifier trained per pathogen. The classifier uses 500 estimators and balanced class weights, trained on one-hot encoded sequences. Negatives were sampled uniformly to match the length distribution of the positive set, and evaluation was performed using 5-fold stratified cross-validation (ROC-AUC). Across pathogens, we obtain strong predictive performance: $AUC = 0.944$ for *E. coli*, $0.942$ for *S. aureus*, $0.913$ for *P. aeruginosa*, $0.905$ for *B. subtilis*, and $0.930$ for *C. albicans*. Then, for a candidate sequence $s$ we compute the predicted antimicrobial-activity probability for each pathogen and aggregate them as $p(s) = \max_{j \in \mathcal{P}} \Pr(y = 1 \mid s, j)$. We convert this score into a log-reward by comparing it to a cutoff $c = 0.95$ in logit space and applying temperature scaling:

$$\log R_{\text{raw}}(s) = \frac{\text{logit}(p(s)) - \text{logit}(c)}{T}, \qquad T = 0.3.$$

If $\log R_{\text{raw}}(s) < 0$, we additionally scale the penalty by the (padded) sequence length $\ell(s)$; finally, we clip $\log R(s)$ to $[-30, 0]$. In particular, sequences with $p(s) \geq c$ satisfy $\log R(s) = 0$, i.e., $R(s) = 1$.

RINGS      8 GAUSSIANS

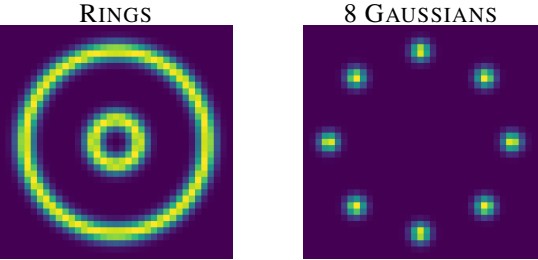

*Figure 11.* **Lazy Random Walk target distributions.**

**Lazy Random Walk Distributions** For the lazy random walk experiments we set $m = 18$ and $T = 2m$ so that its possible to traverse the full domain $[[-m, m]]^d$ within the horizon. We consider two synthetic, multimodal targets and use their (unnormalized) densities as rewards, where $x \in \mathcal{X}$ denotes the terminal position at $t = T$ and we add a small uniform

floor $\lambda$ to avoid zero densities. Concretely, 8 GAUSSIANS is defined as an isotropic mixture of 8 Gaussians whose means are equally spaced on a circle of radius $R = 0.8m$, $\rho_{8G}(x) \propto \sum_{k=1}^{8} \exp\left(-\|x - \mu_k\|_2^2/2\right)$ with $\mu_k = (R\cos\theta_k, R\sin\theta_k)$ and $\theta_k = 2\pi(k-1)/8$; RINGS is a radial mixture over a set of radii $r_1 = 0.2m$ and $r_1 = 0.8m$ with width $\sigma_r = 1$, $\rho_{\text{RINGS}}(x) \propto \sum_i \exp\left(-(\|x\|_2 - r_i)^2/(2\sigma_r^2)\right)$,

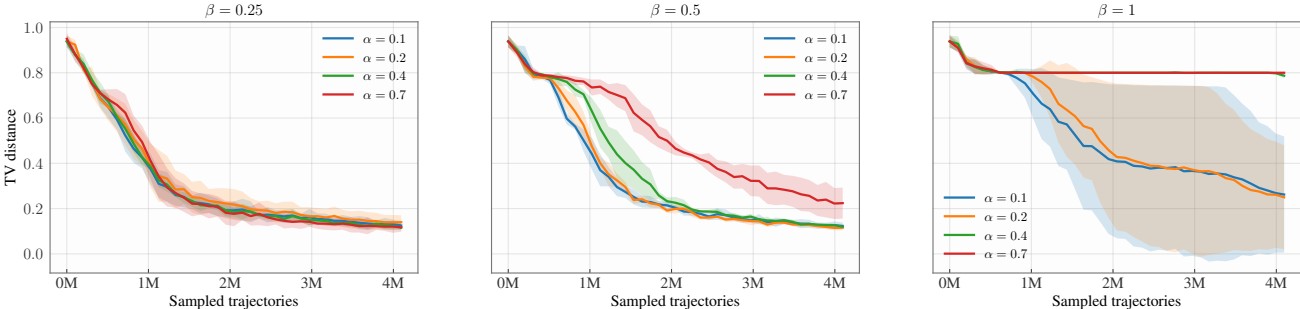

*Figure 12.* **Sensitivity to DTB hyperparameters.** TV distance vs. sampled trajectories on RINGS for a sweep over $\alpha \in \{0.1, 0.2, 0.4, 0.7\}$ (colors) and $\beta \in \{0.25, 0.5, 1\}$ (panels). Solid lines denote the mean across seeds and shaded regions show $\pm 1$ standard deviation. We observe a broad stable regime for $\beta = 0.25$, whereas larger $\beta$ makes training more sensitive to $\alpha$ and can induce failure modes for $\alpha \geq 0.4$ at $\beta = 1$ (TV plateau), together with markedly increased variance.

## D. Hyperparameter Sensitivity

We evaluate the sensitivity of ACE to the DTB hyperparameters $(\alpha, \beta)$ on RINGS, where $\alpha$ sets the allocation threshold used to classify regions as sufficiently learned (and thus excluded from DTB enforcement), while $\beta$ controls reward tempering. We sweep $\alpha \in \{0.1, 0.2, 0.4, 0.7\}$ and $\beta \in \{0.25, 0.5, 1\}$ and report the TV distance as a function of sampled trajectories. Figure 12 shows that performance is stable across a broad range of $\alpha$ for $\beta = 0.25$, while larger $\beta$ increases sensitivity: for $\beta = 0.5$, larger $\alpha$ slows convergence, and for $\beta = 1$ values $\alpha \geq 0.4$ frequently lead to training failure (TV plateaus close to its initial value), with substantially higher variance even for the best-performing settings. These observations motivate our default choice of moderate $\beta$ and small-to-moderate $\alpha$ across environments.

## E. Training ACE in continuous domains

To further highlight ACE's applicability, we show it can be trained in continuous domains using Lahlou et al. (2023)'s framework for GFlowNets in non-countable spaces. For this, we consider $\mathcal{S} = \{(\mathbf{0}, 0)\} \cup (\mathbb{R} \times \{0\} \times \{1\})$ and $\mathcal{X} = \mathbb{R}^2 \times \{2\}$, and let $p_F$ and $p_F^{\triangledown}$ be autoregressive policy functions defined by the transitions $(0, 0) \to (a, 0) \to (a, b)$ for $a, b \in \mathbb{R}$ and

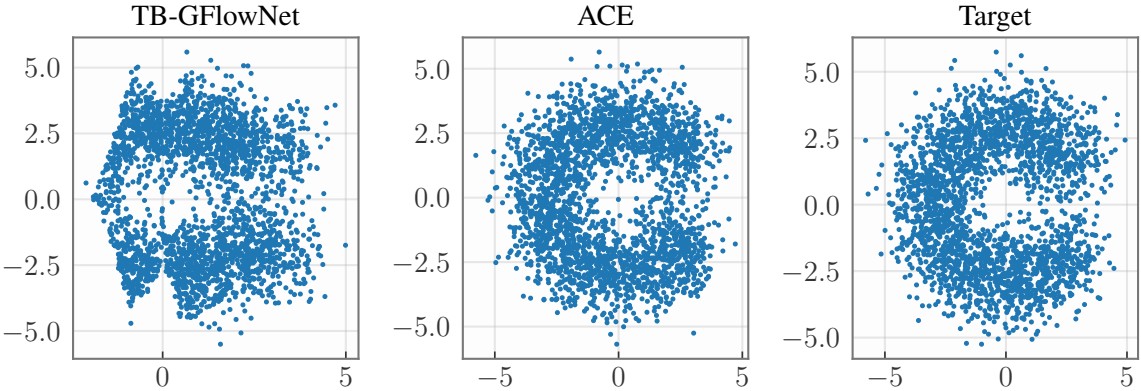

*Figure 13.* ACE accelerates learning convergence in continuous domains.

parameterized as Gaussian mixtures with 10 components, and consider the reward function

$$\log R(x) = \frac{1}{t} \log \sum_{k=1}^{8} \exp\left\{-\frac{|x - \mu_k|^2}{2\sigma^2}\right\}$$

with $\sigma = 0.9$, $t = 1$, and

$$\mu_k = \left(r \cos\theta_k, ; r \sin\theta_k\right), \quad \theta_k = \frac{2\pi k}{9}, \text{ for } k = 1, \ldots, 8 \text{ and } r = 3.$$

We illustrate the outcome of our model, alongside a conventionally trained TB-GFlowNet (as described in (Lahlou et al., 2023)) in Figure 13.

