# OpenReview forum: "Avoid What You Know: Divergent Trajectory Balance for GFlowNets"
_ICML.cc/2026/Conference — ICML 2026 regular_

### Official Review · Reviewer_G2Bj · 2026-03-08

**Soundness:** 3
**Presentation:** 3
**Significance:** 3
**Originality:** 3
**Overall Recommendation:** 4
**Confidence:** 3

**Summary:**

This paper addresses the problem of inefficient exploration in Generative Flow Networks (GFlowNets), where epsilon-greedy exploratory policies tend to over-sample already well-approximated regions while neglecting high-reward but underexplored modes. The authors propose Adaptive Complementary Exploration (ACE), which introduces a second "exploration GFlowNet" trained alongside the canonical GFlowNet. The exploration GFlowNet is governed by a novel *divergent trajectory balance* (DTB) condition that explicitly assigns zero probability to trajectories terminating in over-allocated states, thereby redirecting sampling toward underexplored high-reward regions.

**Compliance With Llm Reviewing Policy:**

Affirmed.

**Key Questions For Authors:**

1. How does ACE perform relative to simply doubling the computational budget of standard TB training (e.g., double batch size or double training iterations)? Is the improvement attributable to the ACE mechanism itself or to the additional model capacity and samples?
2. How sensitive is the method to the quality of both backward policy $p_B$? If $p_B$ is poorly learned (especially early in training), the OA/UA classification could be highly unreliable. Does ACE require a warm-up phase?
3. Definition 3.1 defines OA and UA sets based on a global threshold alpha. In practice, the fraction of states that are over-allocated changes during training. Has any analysis been done on how $|OA(\alpha)| / |X|$ evolves over training, and how this affects learning dynamics?
4. The paper uses beta as a tempering exponent for exploring GFlowNet's reward. What is the interaction between alpha and beta? Could one be expressed as a function of the other to reduce the hyperparameter space?
5. The DTB condition enforces a hard zero on over-allocated trajectories. Would a softer penalty (e.g., down-weighting rather than zeroing) be more robust in practice? The softplus formulation in Equation (9) already provides some smoothing. How critical is this?
6. The DTB loss in Equation (9) includes an indicator function $I[\tau \in OA(\alpha)]$. How is the gradient handled through this indicator? Is the softplus formulation sufficient to ensure well-behaved gradients, or are there dead-gradient issues analogous to ReLU networks?
7. All experiments use TB as the base loss. Does ACE work with other balance conditions (e.g., detailed balance, sub-trajectory balance)? Is there a fundamental reason to prefer TB here?

**Limitations:**

yes

**Strengths And Weaknesses:**

# Strength

1. **Theoretical framework.** The DTB condition is cleanly formulated and yields a well-defined loss function. The theoretical contributions are substantial: Proposition 3.4 (complementary sampling property), Proposition 3.5 (repulsive bound connecting DTB to KL divergences), Proposition 3.8 (gradient pushes exploration toward the complement), and Proposition 3.9 (equilibrium characterization). From my perspective, these are not trivial observations; they provide genuine insight into the algorithm's behavior.

2. **Strong and comprehensive experiments.** The paper evaluates on 7 distinct environments spanning toy problems (lazy random walk, grid world), combinatorial benchmarks (bit sequences, sequence design, bag generation, quadratic knapsack), and a realistic scientific discovery task (AMPs). ACE consistently outperforms all baselines on both mode discovery speed (top-K reward) and distributional accuracy (TV distance). The AMP results (Figure 8) are particularly striking, showing roughly 10x improvement in unique effective peptides.

# Weakness
1. **Computational overhead is not discussed.** ACE trains two GFlowNets simultaneously (canonical $g$ and exploration $g_{\nabla}$), each with its own forward policy, backward policy, and partition function estimate. This roughly doubles the parameter count and per-iteration cost. The paper never quantifies or discusses this overhead.
2. **The OA/UA classification depends on a single-sample estimate of the induced reward.** In Equation (9) and Algorithm 1 (line for determining OA membership), the authors use a single backward sample from $p_B$ to estimate $\hat{R}(x)$ (Remark 3.7). For states where the canonical GFlowNet's induced reward is close to $\alpha  R(x)$, this estimate will be noisy, leading to misclassification. The authors are suggested to discuss the variance of this estimator, consider using multiple samples or an exponential moving average, and empirically assess sensitivity to this approximation.
3. **Proposition 3.9 (Equilibrium State) relies on strong assumptions.** The equilibrium analysis assumes both GFlowNets satisfy their respective TB conditions on *all* trajectories. In practice, with finite-capacity networks and stochastic optimization, this is never achieved. The paper does not discuss how far from equilibrium the practical algorithm operates.
4. **No comparison with replay-buffer-based methods.** Replay buffers (Vemgal et al., 2023) and prioritized experience replay are standard approaches to mitigate mode collapse in GFlowNet training. The paper cites Vemgal et al. (2023) but does not compare against it experimentally. Replay buffers are a simpler mechanism that also helps with exploration; without this comparison, it is unclear how much of ACE's benefit could be achieved with simpler means.

Vemgal, N., Lau, E., and Precup, D. An empirical study of the effectiveness of using a replay buffer on mode discovery in gflownets, 2023.

---

> ### Author Rebuttal · Authors · 2026-03-30
>
> We are thankful for the reviewer’s considerate feedback and support for our work. We below address each of your concerns.
>
> > Computational overhead is not discussed.
>
> ACE has similar complexity and parameter count as AT- and SA-GFlowNets [1,2], our primary baselines. Although wall-clock time is heavily implementation-dependent, we compared ACE vs. TB-GFlowNet trained twice as long on hypergrid and bit tasks. ACE achieves better distributional fit and mode coverage: [hypergrid](https://anonymous.4open.science/r/dtbgflownets-E63C/hypergrids_longer_tb.png), [bits](https://anonymous.4open.science/r/dtbgflownets-E63C/hr_mean_std_bits.png).
>
> > The OA/UA classification depends on a single-sample estimate of the R_hat.
>
> To understand how this affects training, we measured (1) the average variance of R_hat over time and (2) ACE’s performance with more trajectories for R_hat estimation. As expected, variance reduces as training progresses ([plot](https://anonymous.4open.science/r/dtbgflownets-E63C/rhatnoise_hypergrid.png)) . However, using more trajectories does not significantly improve performance ([plot](https://anonymous.4open.science/r/dtbgflownets-E63C/variance_bags.png)). We thus recommend using the single-trajectory estimator.
>
> > Proposition 3.9 relies on strong assumptions.
>
> The central purpose of Proposition 3.9 is to characterize the phase transition occurring at $\alpha = 1$: for $\alpha \le 1$, the exploration GFlowNet focuses on unvisited, high-reward subsets of the state space; for $\alpha > 1$, the stationary distribution corresponds to a tempered reward function $R(x)^{\beta}$. The intended mode-seeking behavior is realized at $\alpha \le 1$, and we set $\alpha \le 1$ in our experiments. We will clarify this connection in the text.
>
> > No comparison with replay-buffer-based methods.
>
> While a replay buffer reinforces already visited high-reward trajectories, ACE guides the GFlowNet towards unvisited regions: they are complementary, not alternatives. Notwithstanding, we ran an experiment comparing a GFlowNet trained with a replay buffer against one trained with ACE. ACE significantly improves convergence speed ([plot](https://anonymous.4open.science/r/dtbgflownets-E63C/hypergrids_replay_buffer.png)).
>
> ---
>
> We are thankful for your questions, which we answer carefully below.
>
> > Model capacity comparison to TB GFlowNet.
>
> We emphasize the neural network’s architecture parameterizing the policy is the same for each considered baseline. As such, their capacities are the same. They also have access to the same number of trajectories during training. This suggests ACE’s consistent superior performance is due to its principled exploration mechanism.
>
> > Does ACE require a warm-up phase? Sensibility to the quality of $p_{B}$?
>
> ACE does not require a warm-up phase. We also ran experiments with an uniform backward policy ([plot](https://anonymous.4open.science/r/dtbgflownets-E63C/hypergrids_uniform_policies.png), [plot](https://anonymous.4open.science/r/dtbgflownets-E63C/bags_uniform_policies.png)) for the 16x16 hypergrid environment. ACE still leads to faster learning convergence. Importantly, for autoregressive modeling (e.g., sequence generation), the optimal $p_{B}$ is known; thus, OA and UA are exactly computed.
>
> > OA changes during training. Evolution of OA($\alpha$) / |X| throughout training?
>
> As in [1, 2], ACE is trained under a non-stationary reward. Our empirical analysis suggested this does not negatively affect training. Additionally, we measured $\mathrm{OA}(\alpha) / |X|$ during training for the 16x16 hypergrid task. As expected, in lieu of Proposition 3.9, OA($\alpha$) / |X| converges to 1 for $\alpha < 1$, and to 0 for $\alpha > 1$ ([plot](https://anonymous.4open.science/r/dtbgflownets-E63C/oa_curves.png)).
>
> > Can $\alpha$ be written as a function of $\beta$?
>
> There is no clear relationship between $\alpha$ and $\beta$, which can be chosen independently. It is worth noticing, however, that ACE has fewer hyperparameters than both AT- and SA-GFlowNets.
>
> > Can we use a softer penalty (instead of an indicator function)?
>
> In early experiments, we substituted the indicator function with a softer alternative, without noticeable empirical benefits. Our method, in contrast, can be principledly derived from the DTB condition. Also, as $\mathrm{OA}(\alpha)$ depends only on the canonical GFlowNet, and not on the exploration GFlowNet, we do not compute gradients through it when minimizing the DTB loss.
>
> > Can ACE be derived using other balance conditions?
>
> The main reason we chose TB is for its widespread adoption in the literature. However, the canonical GFlowNet in ACE can be trained with any loss function providing an estimate of the log-partition function (e.g., DB, SubTB).
>
> [1] Kim et al., ICLR 2025c.
>
> [2] Madan et al., ICLR 2025.
>
> [3] Malkin et al., NeurIPS 2022.
>
> [4] Bengio et al., NeurIPS 2021.
>
> ---
> We are thankful for your valuable input! We will update the manuscript accordingly.

---

> > ### Author Rebuttal · Reviewer_G2Bj · 2026-04-02
> >
> > I acknowledge that my concerns are addressed by the authors. Thanks to the author for putting effort into this rebuttal. I will maintain my score.

---

### Official Review · Reviewer_TSkN · 2026-03-11

**Soundness:** 3
**Presentation:** 2
**Significance:** 2
**Originality:** 3
**Overall Recommendation:** 4
**Confidence:** 3

**Summary:**

This paper proposes Adaptive Complementary Exploration (ACE) for GFlowNets. The main idea is to train an additional exploration policy that explicitly targets high-reward but under-explored regions, rather than continuing to spend samples on regions that the canonical GFlowNet already covers well. The paper supports this idea with a trajectory-balance-style objective, several theoretical characterizations, and experiments on multiple GFlowNet benchmarks.

**Compliance With Llm Reviewing Policy:**

Affirmed.

**Final Justification:**

The rebuttal addressed my main concerns and improved my overall assessment of the paper.

**Key Questions For Authors:**

See "Weaknesses".

**Limitations:**

See "Weaknesses".

**Strengths And Weaknesses:**

**Strengths**
1. The motivation is clear: standard exploration strategies can waste samples on already well-covered regions, whereas ACE is designed to focus exploration on under-allocated but valuable parts of the state space. This is a natural and practically relevant idea.
2. Rather than just adding noise or an intrinsic reward bonus, the paper introduces a separate exploratory policy and analyzes its behavior through several propositions about support, collapse avoidance, and equilibrium behavior. Even if some assumptions are strong, the framework is more principled than a purely heuristic exploration tweak.

**Weaknesses**
1. The proposition assumes zero loss for all trajectories and then derives that the exploratory terminal distribution is proportional to $R(x)^{\beta}$ on the under-allocated set. While this is a clean representation result, it relies on an exact optimization condition that is much stronger than what the actual stochastic training procedure can guarantee. The paper does not provide an approximation result connecting small empirical loss to closeness of the learned exploration distribution to this ideal form.
2. This result assumes that both the canonical and exploratory GFlowNets have already collapsed to the same subset $C$, and that trajectory-balance conditions hold on all trajectories reaching $C$. Under these assumptions, the proposition shows that the on-policy gradient pushes the exploratory model away from $C$. This supports the intuition that ACE is repulsive with respect to collapsed modes, but the setting is fairly stylized. It does not address more realistic cases such as partial collapse, approximate balance, or the two models collapsing to different regions.
3. The proposition gives a neat equilibrium description under mutual best-response assumptions, including different regimes depending on $\alpha$. However, ACE is trained via an alternating two-model procedure, and the paper does not really study whether this coupled optimization is stable, convergent, or prone to oscillation in practice. So the static equilibrium analysis is useful, but it leaves a substantial gap between the formal characterization and the actual training dynamics.
4. Although the paper evaluates on several benchmarks, many of them are still standard or synthetic environments. Even the AMP design task relies on a surrogate reward model rather than a stronger real-world validation pipeline. As a result, the evidence that ACE will remain effective in more complex, noisy, or genuinely high-stakes scientific discovery settings is still somewhat limited.

---

> ### Author Rebuttal · Authors · 2026-03-30
>
> We are grateful for the reviewer’s careful feedback. We will gladly incorporate the additional discussions into the updated text.
>
> > Approximation results for small losses.
>
> Thank you for the opportunity to strengthen our theoretical contributions. Although associating empirical losses to model behavior is primarily a question of generalization, for which existing theory cannot be directly applied to ACE, our algorithm’s behavior for an $\epsilon$-approximate solution can be cleanly characterized. To see this, we assume that, for a given $\epsilon > 0$,
>
> $$
> \left( \log \left( \frac{Z p_{F}^{\nabla}(\tau)}{R(x)^{\beta} p_{B}^{\nabla}(\tau | x)}  + \mathbf{I}[\tau \in \mathrm{OA}(\alpha)] \right) \right)^{2} \le \epsilon
> $$
>
> for every trajectory $\tau$. Under these conditions, it can be shown the marginal distribution $p_{\top}^{\nabla}$ of the exploration GFlowNet satisfies, for $\kappa(\epsilon) = \exp\{\sqrt{\epsilon}\}$,
>
> $$
> 	0 \le p_{\top}^{\nabla}(x) \le \frac{R(x)^{\beta}}{Z_{\nabla}} \cdot \left( \kappa (\epsilon) - 1 \right)
> $$
>
> for $x \in \mathrm{OA}(\alpha)$ and
> $$
>  	\frac{R(x)^{\beta}}{Z_{\nabla}} \kappa(\epsilon)^{-1}  \le p_{\top}^{\nabla}(x) \le  \frac{R(x)^{\beta}}{Z_{\nabla}} \kappa(\epsilon)
> $$
>
> otherwise. We will include formal derivations for these inequalities (omitted due to space constraints) in the revised manuscript.
>
> This shows the exploration GFlowNet’s claimed behavior of assigning negligible probability mass to over-allocated regions of the state space is preserved under an $\epsilon$-approximate minimization of the loss function. Additionally, our results also provide guidance in choosing $\alpha$ and $\beta$, e.g., letting $\beta < 1$ for sparse, multi-modal distributions.
>
> > What happens when models are collapsed in different subsets?
>
> This is an interesting problem. We consider the case in which the exploration and canonical GFlowNets are collapsed in a different subset of the state space, i.e., $C$ and $C_{\nabla}$, and let $\theta$ be the corresponding parameters of the exploration GFlowNet’s forward policy ($p_{F}^{\nabla}$). Then, through the same techniques used in the proof of Proposition 3.8, we verify that
> $$
> \mathbb{E}\_{\tau \sim p\_{F}^{\nabla}} \nabla\_{\theta} \mathcal{L}\_{\nabla}(\tau ; \theta)  = - ( \log 2 ) \nabla_{\theta} \sum\_{\tau \in C^{c} \cup C\_{\nabla}^{c}} p\_{F}^{\nabla}(\tau ; \theta).
> $$
>
> In particular, when $C_{\nabla} \supseteq C$, i.e., the exploration GFlowNet is collapsed on a superset of $C$, we recover our original statement. More broadly, this equation states that the gradient of our DTB loss function steers the model towards under-allocated regions by the canonical GFlowNet, represented by $C^{c}$ ($C$'s complement), _and_ guides the exploration GFlowNet away from collapse, via $C_{\nabla}^{c}$.
>
> > the paper does not really study whether this coupled optimization is stable, convergent, or prone to oscillation in practice.
>
> We agree that Algorithm 1 is not guaranteed to converge to the stated equilibrium in Proposition 3.9, whose main purpose is to characterize the phase-transition that occurs at $\alpha = 1$. The statement highlights that, when $\alpha > 1$, the exploration GFlowNet may over-emphasize well-learned regions, as its equilibrium distribution is supported on the entire state space. When $\alpha \le 1$, in contrast, the equilibrium distribution indicates the exploratory policy focuses primarily on unvisited subsets of the state space, which is the desired behavior. This is the reason we set $\alpha \le 1$ in our experiments. We will make this connection clearer in the manuscript.
>
> Empirically, we did not find ACE to exhibit training instability. To see this, we plotted the learning curve of both ACE’s canonical GFlowNet and a standard TB GFlowNet for the task of bag generation (same setting as in Figure 4(a)). Results [here](https://anonymous.4open.science/r/dtbgflownets-E63C/bags_learning_curve.png). Notably, neither method exhibits significant training instability.
>
> > Although the paper evaluates on several benchmarks, many of them are still standard or synthetic environments.
>
> We are glad to further elaborate on our experiments. Importantly, our empirical analysis is aligned with the broader GFlowNet literature, both in scale and diversity [1, 2]. The use of a proxy network for the reward function, in particular, is a standard technique for training GFlowNets in molecule generation tasks [1, 2].
>
> Nonetheless, we would be happy to discuss additional potential applications for ACE.
>
> [1] Adaptive teachers for amortized samplers, Kim et al., ICLR 2025.
>
> [2] Towards Improving Exploration through Sibling Augmented GFlowNets, Madan et al., ICLR 2025.
>
> ---
> Many thanks for the careful review and insightful questions! Their inclusion will significantly strengthen our work’s contributions.

---

> > ### Author Rebuttal · Reviewer_TSkN · 2026-04-03
> >
> > Thank the authors for the rebuttal. My concerns have been adequately addressed. I'd like to raise my score.

---

### Official Review · Reviewer_thzQ · 2026-03-12

**Soundness:** 3
**Presentation:** 3
**Significance:** 3
**Originality:** 3
**Overall Recommendation:** 4
**Confidence:** 3

**Summary:**

this paper proposes a method called ACE to improve exploration in gflownet training. the main idea is to introduce another objective called DTB and train a separate exploration gflownet that focuses on high reward states the main model might be missing.

the two networks basically work together. the exploration one tries to cover regions that havent been explored much, and the canonical one learns from trajectories produced by both. the paper also gives some theoretical discussion about the equilibrium behavior and reports strong empirical results on several benchmarks like bit sequences, peptide design and combinatorial optimization tasks.

**Compliance With Llm Reviewing Policy:**

Affirmed.

**Final Justification:**

The new experiments on Q3 are convincing, so I am willing the raise the score.

**Key Questions For Authors:**

figure 10 shows that beta = 1 with alpha ≥ 0.4 can basically lead to training failure. is there some principled way to choose these hyperparameters, or do people just have to tune them depending on the environment? this feels like it could be a practical issue for users.

the batch size for ACE and SA gflownets is set to half of the other baselines for fairness. but that also means ACE is doing more computation per effective gradient step. so it would be helpful if the authors could clarify whether the comparison is really about sample efficiency or more about compute efficiency.

in algorithm 1 the over allocation check uses only a single backward sample to decide whether a trajectory is in OA(alpha). how sensitive is this to noise in the estimate, especially early in training when the backward policy is still poorly calibrated?

**Limitations:**

it might also help to briefly mention a few practical things. for example the extra compute cost of running two gflownets at the same time, how noisy the over and under allocation classification might be early in training since it relies on a single sample estimate of Rhat, and how the method might behave in very large state spaces or when reward evaluations are expensive. the AMP results are interesting but they are still on fairly limited sequence lengths.

**Strengths And Weaknesses:**

strengths

the way the paper frames exploration as a complementary balance condition is actually pretty nice and clean. compared with earlier curiosity style methods that try to encourage novelty through proxy rewards, this one more directly pushes the model to avoid regions it already knows well. that feels like a different and more principled angle.

the empirical results also look quite strong and cover several different types of tasks. ACE beats solid baselines pretty consistently. the AMP experiment is especially interesting since it shows a big jump in the number of unique peptides discovered.

weaknesses

the method adds two extra hyperparameters, alpha and beta, and they seem to matter quite a lot. figure 10 even shows that large beta values can basically break training, and the paper mostly just suggests not setting it too large. some clearer guidance on how to pick these would make the method easier to use.

the theory part mostly describes the equilibrium behavior under fairly ideal assumptions like exact TB satisfaction or collapsed policies. it doesnt really say much about what happens during the actual training dynamics, which is probably where most of the interesting behavior happens.

---

> ### Author Rebuttal · Authors · 2026-03-30
>
> We are grateful for the reviewer’s thoughtful feedback and contributions towards strengthening our work. We will update the manuscript with further discussions regarding the choice of hyperparameters, an analysis of ACE’s computational complexity, and the sensitivity of our results to the accuracy of $\hat{R}_{\mathfrak{g}}$’s estimate, as elaborated below.
>
> > On the choice of $\alpha$ and $\beta$.
>
> We appreciate the opportunity to discuss the hyperparameter selection for ACE. Simply put, $\beta$ influences the _flatness_ of the reward function for the exploration network. By selecting a $\beta$ that is too large, the induced equilibrium reward function (per Proposition 3.4) is peaky, and exploration is harder. As learning requires effective mode visitation, we set $\beta = 0.25$ for all but the AMP task, for which we let $\beta = 1$. Regarding $\alpha$, we found $\alpha \in [0.2, 0.3]$ to work well.
>
> On top of that, ACE requires fewer hyperparameters than both SA-GFlowNets (five:  hyperparameters, $\beta_{e_{BN}}, \beta_{e_{SN}}, \beta_{i}, \beta_{BN}, \text{ and } \beta_{SN}$) and AT-GFlowNets (three: $\alpha$, $C$, and the proportion of Teacher/Student/Replay buffer samples per gradient step), which are our primary baselines.
>
> > On the assumptions of our theoretical results.
>
> We are glad to elaborate further on our theoretical results.
>
> We would first like to note that the objective of our analysis is to understand the behavior of ACE at equilibrium, i.e., when the system is either collapsed (Proposition 3.8) or perfectly trained (Propositions 3.4, 3.9). This provides a clearer understanding of the influence of $\alpha$ and $\beta$ on training, as discussed. It also underlines how ACE provably avoids certain catastrophic failure modes in GFlowNet training, which is a unique feature of our algorithm.
>
> While we agree that an in-depth analysis of ACE’s training dynamics would provide a clearer picture of its behavior, we also emphasize that this is not feasible within existing frameworks for Deep Learning theory. ACE is trained with adaptive stochastic gradient estimators (Adam) over non-stationary and dynamically generated data (trajectories) drawn from a combinatorial space, the analysis of which significantly exceeds the reach of existing mathematical tools for deep learning. In our opinion, this merits a work of its own.
>
> > it would be helpful if the authors could clarify whether the comparison is really about sample efficiency or more about compute efficiency.
>
> The comparison is about sample efficiency, i.e., how many states should be observed to achieve a certain performance threshold (e.g., mode exploration, distributional accuracy). This is the standard metric upon which prior works rely to assess their proposed methods (e.g., [1, 2, 3]). The AT-GFlowNet paper [2], for instance, refers to the number of “reward queries” needed for training, while the TB paper [1] measures learning convergence as a function of the number of visited states. We will emphasize this in the revised manuscript.
>
> [1] Trajectory balance: Improved credit assignment in GFlowNets, Malkin et al., NeurIPS 2022.
>
> [2] Adaptive teachers for amortized samplers, Kim et al., ICLR 2025.
>
> [3] Towards Improving Exploration through Sibling Augmented GFlowNets, Madan et al., ICLR 2025.
>
> > how sensitive is this to noise in the estimate, especially early in training when the backward policy is still poorly calibrated?
>
> Thank you for the question. To understand this, we (1) measured the empirical standard deviation of $\hat{R}\_{\mathfrak{g}}$ as training progresses (averaged over all terminal states) and (2) verified ACE’s performance as the number of trajectories for estimating $\hat{R}\_{\mathfrak{g}}$ increases.
>
> Results for (1) and (2) can be accessed [here](https://anonymous.4open.science/r/dtbgflownets-E63C/rhatnoise_hypergrid.png) and [here](https://anonymous.4open.science/r/dtbgflownets-E63C/variance_bags.png), respectively. As expected, the standard deviation of our estimator for $\hat{R}_{\mathfrak{g}}$ decreases as training progresses. However, our experiments suggested that the increased computational cost does not always lead to a statistically significant speed up in learning convergence or state space exploration for the tasks we considered. In practice, we thus recommend using a single trajectory for estimating $\hat{R}\_{\mathfrak{g}}$.
>
> Importantly, for autoregressive modelling (e.g., the AMP task), there is a only one trajectory leading to each state. Hence, both OA and UA are exactly computed.
>
> > Limitations
>
> Thank you for pointing these out. We will include a detailed discussion about these aspects in the revised manuscript, as well as about the above experiments. In particular, we will emphasize that, when reward evaluation is costly, ACE’s remarkable sample efficiency is important for reducing the number of reward queries during training.
>
> ---
> Thank you for the careful feedback and suggestions!

---

> > ### Author Rebuttal · Reviewer_thzQ · 2026-04-03
> >
> > Thank you for the rebuttal. The new experiments on Q3 are convincing.The clarification on Q2 is also appreciated.
> > The concern on training dynamics (W2) remains, but I understand this is a known limitation of current theory. I am willing to raise my score.

---

### Official Review · Reviewer_1y3n · 2026-03-12

**Soundness:** 3
**Presentation:** 3
**Significance:** 4
**Originality:** 4
**Overall Recommendation:** 5
**Confidence:** 4

**Summary:**

The paper introduces Adaptive Complementary Exploration (ACE), a method designed to overcome the exploration bottlenecks in GFlowNets. The core innovation is the Divergent Trajectory Balance (DTB) objective. ACE operates as a two-player system: a "canonical" GFlowNet learns the target distribution, while an "exploration" GFlowNet is explicitly trained to sample from under-allocated regions. DTB works by assigning zero probability to over-allocated regions. The authors provide theoretical guarantees regarding the equilibrium of this system and demonstrate, through extensive experiments that ACE significantly accelerates mode discovery and improves approximation accuracy compared to existing baselines.

**Compliance With Llm Reviewing Policy:**

Affirmed.

**Final Justification:**

The authors addressed all my questions, so I maintain my positive review

**Key Questions For Authors:**

- The paper focuses on discrete GFlowNets, but would this method be applicable to continuous settings?
- As mentioned above, the single sample used to estimate $\hat{R}_{\mathfrak{g}}(x)$ is extremely stochastic. It would be great to A estimate the uncertainty in $\hat{R}_{\mathfrak{g}}(x)$ for a fixed policy for some of the examples, and also to ablate what happens when more samples are used.
- Figure 10 shows that for $\beta=1$, the model is very sensitive to $\alpha$. In a real-world scientific discovery task where the true distribution is unknown, how do you recommend $\alpha$ is selected?

**Limitations:**

Yes

**Strengths And Weaknesses:**

**Soundness**: The paper is theoretically rigorous. The authors provide proofs for their various claims. Empirical evaluation is comprehensive, and experiments are through and well explained. The only weakness is that the estimation of the induced reward $\hat{R}_{\mathfrak{g}}(x)$ relies on a single sample from the backward policy $p_B$, which can be extremely noisy. It would be interesting to see an ablation with increasing number of samples.

**Presentation**: The writing is clear, and the flow and structure is good. The diagrams are helpful, though a more introductory diagram for the less expert reader would help.

**Significance**: Exploration is an important problem in GFlowNets, and a key bottleneck for training in practical applications. The experiments show the method working in high dimensional and real problems.

**Originality**: While other works have recently tried to address this problem, the approach from the authors is novel and differentiated.

---

> ### Author Rebuttal · Authors · 2026-03-30
>
> We appreciate the reviewer’s supportive review of our work. We will incorporate all additional experiments and discussions below into the revised manuscript.
>
>  > The paper focuses on discrete GFlowNets, but would this method be applicable to continuous settings?
>
> Thank you for the question. In principle, extending our method (ACE) to continuous domains is certainly possible. For this, both the policy parameterization and the theoretical formalism would have to be modified.
>
> 1. As a softmax neural network is no longer applicable in this setting, we could follow [1] and learn instead the mean, variance, and weights of a Gaussian mixture representing the forward policy $p_{F}(\cdot | s)$.
> 2. As in [1], a few technical changes, such as replacing the DTB condition in Definition 3.2 by a Kolmogorov-style almost sure equality, would be needed.
>
> To illustrate this, we have included an experiment in which we parameterize a GFlowNet as in 1. above and learn to sample from a sparse Gaussian mixture over a 2-dimensional domain, a standard task in the continuous GFlowNet literature (e.g., [1]). Results may be consulted in this [anonymized URL](https://anonymous.4open.science/r/dtbgflownets-E63C/gaussian.png). As we can observe, ACE correctly captures the target distribution, while achieving a better goodness of fit than a GFlowNet trained only by minimizing the TB loss. In particular, ACE attains a 2-Wasserstein distance of $0.22$ to the target, and the TB GFlowNet, of $1.62$.
>
> With this in mind, an in-depth investigation of ACE in continuous domain is an interesting direction.
>
> [1] A Theory of Continuous Generative Flow Networks, Lahlou et al. ICML 2023.
>
> > It would be great to estimate the uncertainty in $\hat{R}\_{\mathfrak{g}}(x)$ for a fixed policy for some of the examples
>
> We appreciate your suggestion. In fact, ACE imposes a trade-off between computational and statistical efficiency when estimating $\hat{R}_{\mathfrak{g}}$, with a larger number of trajectories providing a better estimate at a larger cost.
>
> To understand this, we ran an experiment in which we measure the average standard deviation of $\hat{R}_{\mathfrak{g}}(x)$ across $x \in \mathcal{X}$ throughout training for the 16x16 hypergrid task, for which we can directly enumerate the (otherwise combinatorially intractable) state space $\mathcal{X}$. Results can be found [here](https://anonymous.4open.science/r/dtbgflownets-E63C/rhatnoise_hypergrid.png). As expected, the estimator’s variance decreases as training progresses. (In all experiments, we computed standard deviations from 3 independent runs).
>
> > and also to ablate what happens when more samples are used.
>
> Additionally, we measured ACE's performance for the bag generation and hypergrid tasks (Section 4) as we increase the number of trajectories for estimating $\hat{R}\_{\mathfrak{g}}$, and have not observed a statistically significant improvement in state space exploration. We share the results [in this URL](https://anonymous.4open.science/r/dtbgflownets-E63C/variance_bags.png).
>
> From a practical viewpoint, therefore, we recommend using a single backward trajectory for estimating $\hat{R}_{\mathfrak{g}}$, which we found to consistently improve upon existing baselines in all experiments.
>
> It should also be noted that, for autoregressive modelling (such as our AMP task), there is a single trajectory corresponding to each state, and the stochasticity of the estimate is consequently not an issue.
>
> > How do you recommend selectin $\alpha$?
>
> We acknowledge the importance of properly choosing the hyperparameters for our algorithm.
>
> Our experiments (including Figure 10) suggest that, unless there is evidence supporting the choice for $\beta \ge 1$, such as a relatively flat reward function, a practitioner should set $\beta < 1$. We also suggest setting $\alpha < 1$. In particular, our empirical analysis found $\beta = 0.25$ and $\alpha = 0.3$ to be consistently effective for most experiments, and we recommend these as default hyperparameters.
>
> Importantly, ACE requires fewer hyperparameters than both SA-GFlowNet, which has 5 hyperparameters, $\beta_{e_{BN}}, \beta_{e_{SN}}, \beta_{i}, \beta_{BN}, \text{ and } \beta_{SN}$, and AT-GFlowNet, which has $\alpha$, $C$, and the proportion of Teacher/Student/Replay buffer samplers at each gradient step as hyperparameters.
>
> Standard hyperparameter selection techniques can also be applied in this setting. Since ACE involves a low-dimensional hyperparameter space, one practical approach is to run a small number of short pilot trainings with different hyperparameter configurations and then select the one with the best validation performance.
>
> ---
>
> We are glad for your suggestions and support for our work! We will update the manuscript with the novel experiments and discussions.

---

> > ### Author Rebuttal · Reviewer_1y3n · 2026-04-08
> >
> > Overall, the rebuttal strengthens my confidence in the paper. The authors have addressed my questions satisfactorily, and I remain supportive of acceptance.

---

### Decision · Program_Chairs · 2026-04-30

**Decision:**

Accept (regular)

**Comment:**

This paper proposes Divergent Trajectory Balance (DTB) to address exploration bottlenecks in GFlowNets by training an exploration network that targets underexplored high-reward regions while the canonical network learns the target distribution. Reviews: 1 Accept (5, confidence 4) + 3 Weak Accept (4, confidence 3 each). Strengths: novel DTB mechanism, strong theory with 4 propositions, comprehensive experiments across 7 benchmarks (AMP showing big improvement), responsive rebuttal. All concerns were addressed.  Interesting paper!  So I suggest: "ACCEPT".